# ROBUST FINETUNING OF VISION-LANGUAGE-ACTION ROBOT POLICIES VIA PARAMETER MERGING

**Yajat Yadav**\*, **Zhiyuan Zhou**\*, **Andrew Wagenmaker, Karl Pertsch, Sergey Levine**
UC Berkeley

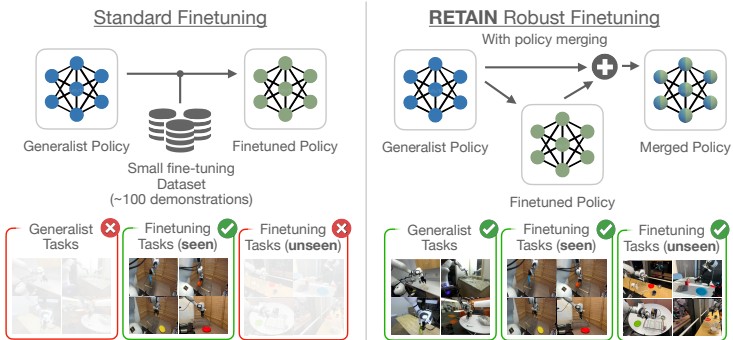

Figure 1: Naive approaches for finetuning of generalist policies narrowly improve target task performance on settings seen in the finetuning data, but fail to generalize or retain generality beyond the target task. We propose a simple solution: by averaging the generalist policy before and after finetuning, *in weight space*, we obtain finetuned policies that (1) significantly improve generalization ability to unseen variations of the target task, and (2) retain generalist capabilities on non-target tasks. Our approach RETAIN is a simple solution for robust policy finetuning.

## ABSTRACT

Generalist robot policies, trained on large and diverse datasets, have demonstrated the ability to generalize across a wide spectrum of behaviors, enabling a single policy to act in varied real-world environments. However, they still fall short on new tasks not covered in the training data. When finetuned on limited demonstrations of a new task, these policies often overfit to the specific demonstrations—not only losing their prior abilities to solve a wide variety of generalist tasks but also failing to generalize within the new task itself. In this work, we aim to develop a method that preserves the generalization capabilities of the generalist policy during finetuning, allowing a single policy to robustly incorporate a new skill into its repertoire. Our goal is a *single* policy that both learns to generalize to variations of the new task and retains the broad competencies gained from pretraining. We show that this can be achieved through a simple yet effective strategy: interpolating the weights of a finetuned model with that of the pretrained model. We show, across extensive simulated and real-world experiments, that such *model merging* produces a single model that inherits the generalist abilities of the base model and learns to solve the new task robustly, outperforming both the pretrained and finetuned model on out-of-distribution variations of the new task. Moreover, we show that model merging performance scales with the amount of pretraining data, and enables continual acquisition of new skills in a lifelong learning setting, without sacrificing previously learned generalist abilities.

## 1 INTRODUCTION

Generalist robot policies trained on large corpora of data have recently shown impressive generalization abilities: out of the box, they can perform a range of tasks in unseen environments, generalize across scenes, viewpoints, objects, and language instructions (Intelligence et al.; Kim et al., 2024;

---

\* Core contributors. Correspondence to yajatyadav@berkeley.edu and zhiyuan_zhou@berkeley.edu. Project page at https://retain.yajatyadav.com

Team et al., 2025; NVIDIA et al., 2025; Liu et al., 2024b; Qu et al., 2025; Gao et al., 2025; Amin et al., 2025). Though impressively general, these generalist policies often need to be adapted to perform effectively on downstream tasks or a new robot system, which is most commonly achieved by finetuning them on a curated dataset of demonstrations for the target task. While prior work has shown that such finetuning can lead to robust policies with tens or hundreds of hours of finetuning data (Black et al., 2024; Intelligence et al.; Bousmalis et al., 2023; Brohan et al., 2023), collecting such amounts of robot demonstration data is challenging. As a result, in practice often less than 100 demonstrations or a few hours of robot data are used for finetuning (Kim et al., 2024; Team et al., 2024; Kim et al., 2025). Existing approaches for robot policy finetuning struggle to *preserve* the generality of the pretrained model in such low-data regimes, and fail to robustly generalize far beyond the exact viewpoints, objects, and scenarios seen in the finetuning data (Gao et al., 2025; Xiong et al., 2020; Zhang et al., 2025; Wang, 2025; Xiang et al., 2025; Zhu et al., 2025; Kaplanis et al., 2019). To expand the usability of generalist policies, we need *robust* finetuning approaches that better preserve the generality of pretrained robot policies and allow us to generalize to a broader set of scenarios *on the target task*.

In this work, we introduce RETAIN (**R**obust fin**E**-tuning wi**T**h p**A**rameter merg**IN**g), a surprisingly simple approach for robust robot policy finetuning. By simply interpolating the *weights* of the pretrained generalist policy before and after finetuning on the target task (see Fig. 1), we obtain checkpoints that match the performance of the finetuned policy on scenarios present in the finetuning data, while generalizing significantly better to unseen variations of the target task, such as unseen object instances, positions, or viewpoints. Additionally, we observe that RETAIN preserves the generalist capabilities of the pretrained policy also on tasks *other than the target task*, allowing us to use RETAIN in a continual learning setup by sequentially *merging* new skills into pretrained generalist policies (in a literal sense). We demonstrate the effectiveness of RETAIN for robust policy finetuning and sequential skill acquisition across a range of real-world and simulated finetuning tasks, achieving state-of-the-art finetuning performance. We also show that RETAIN gets even more effective when the pretrained policy is trained on more data. While previous work has investigated interpolating model weights of pretrained and fine-tuned models for vision and language (Wortsman et al., 2022b;a; Ilharco et al., 2022), this is to our knowledge the first work to investigate and analyze parameter merging for robot policies and use it to enable continual acquisition of new robotic skills.

In summary, our contributions are threefold: (1) we introduce a simple approach for robust robot policy finetuning via policy parameter merging, (2) we extensively evaluate our approach across real-world and simulated robot tasks, and analyze which factors enable successful policy merging, (3) we demonstrate that our approach enables continual merging of new robot skills into state-of-the-art generalist policies. Policies finetuned with our method generalize to novel scenarios for the new skill on real robots with $\sim 40\%$ higher success rate on average than best prior finetuning methods.

## 2 RELATED WORK

**Adapting generalist policies on new tasks.** Fueled by large-scale human teleoperated robot datasets (Collaboration et al., 2023; Walke et al., 2023; Khazatsky et al., 2024; Shah et al., 2023), generalist robot policies have recently solved a wide range of tasks across diverse scenes (Black et al., 2024; Team et al., 2025; NVIDIA et al., 2025; Kim et al., 2024; Brohan et al., 2023; Liu et al., 2024b; Anil et al., 2023; Sridhar et al., 2024; Qu et al., 2025). Yet, even state-of-the-art generalist policies typically need to be adapted for any given target task to achieve high performance (Black et al., 2024). Thus, a number of approaches have been proposed for training generalist policies on a new target task: from simply finetuning them on a dataset of target task demonstrations (Black et al., 2024; Kim et al.,

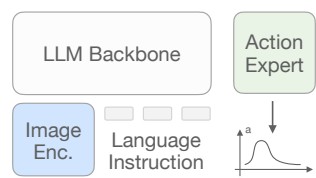

Figure 2: State-of-the-art generalist policies typically consist of a vision encoder, language model backbone, and action "expert" (decoder).

2024; Team et al., 2025), or mixing the outputs of the pretrained and fine-tuned policies (Bagatella et al., 2025), to alternative approaches like online and offline reinforcement learning (Mark et al., 2024; Huang et al., 2025; Yang et al., 2023b; Hu et al., 2024; Zhou et al., 2024; Xu et al., 2024), retrieval-based adaptation (Long et al., 2025; Di Palo & Johns, 2024) or in-context improvement (Fu et al., 2024a; Sridhar et al., 2025). In this work, we focus on the most common setting, in which policies are finetuned on a target task using a small dataset of demonstrations. Various finetuning approaches have been proposed in the literature, from simply adapting the full network on the target

dataset (Kim et al., 2024; Black et al., 2024), to mixing target and pretraining data (Bousmalis et al., 2023; Fu et al., 2024b; Dass et al., 2025), or freezing parts of the network during finetuning (Liu et al., 2023; Liang et al., 2022; Sharma et al., 2023; Xu et al., 2023). While such approaches may prove effective for learning robust target task policies in "large-data" finetuning regimes with tens to hundreds of hours of finetuning data (Black et al., 2023; Intelligence et al.; Bousmalis et al., 2023; Brohan et al., 2023), they often struggle to retain the generality of the pretrained policy in more common, accessible settings with 100 or less target task demonstrations. In such scenarios, finetuned policies often struggle to generalize meaningfully beyond the conditions seen in the finetuning dataset (Zhang et al., 2025), even if the base policy had broad generalization capabilities. In this work, we propose a simple alternative for robust policy finetuning in low-data regimes. Instead of directly using the finetuned policy, we observe that merging the pretrained and finetuned policy checkpoints *in weight space* leads to significantly improved generalization on target tasks at no additional training or inference cost.

**Model parameter merging.** Our approach is inspired by work on model weight merging in vision and language domains (Wang et al., 2024b; Yadav et al., 2023; 2024; Nasery et al., 2025; Lu et al., 2025; Jang et al., 2024; Matena & Raffel, 2022; Yang et al., 2023a; Jin et al., 2022). These works demonstrate that interpolating between the weights of multiple finetuned models, or between pretrained and finetuned models, can combine their capabilities or make them more robust to distribution shifts (Wortsman et al., 2022b;a; Ilharco et al., 2022; Neyshabur et al., 2020; Marincione et al., 2025). To our knowledge, our work is the first to demonstrate the effectiveness of model merging in the context of generalist robot policies, and combining it with co-training to further improve upon vanilla model merging. Additionally, we analyze the importance of the generality of the base model, as well as the importance of different parameter groups in vision-language-action (VLA) policies and find it often sufficient to only merge parameters from the language model backbone.

**Continual learning.** The focus of our work is on improving generalization of finetuned policies on a target task. However, in addition, we find that our model merging approach is also effective at retaining the generalist policy's performance on tasks from the pretraining distribution. As such, we demonstrate that it can be used to sequentially merge multiple skills into a single pretrained policy checkpoint while retaining generality. This setting is typically referred to as *continual learning* and there is a large body of literature, both outside (Kirkpatrick et al., 2017; Schwarz et al., 2018; Lopez-Paz & Ranzato, 2017; Wang et al., 2024c) and within robotics (Lesort et al., 2020; Auddy et al., 2023; Wan et al., 2024; Meng et al., 2025; Liu et al., 2024a; Wołczyk et al., 2021). Our work differs from this line of research in that we aim to inherit and pass on the generalization ability of a pretrained model to learn new tasks robustly, whereas continual learning methods generally focus on not forgetting old skills seen during the agent's lifetime.

## 3 PROBLEM SETTING

The goal of our work is to develop an approach for *robust* policy finetuning, in which a generalist policy is finetuned to a new target task and generalizes to *unseen variations of that target task*, like new object instances, viewpoints, scenes, or lighting conditions, while also preserving its generalist abilities. Formally, let $\mathcal{M}$ denote an environment, $\mathcal{S}$ denote observations (e.g., images, proprioception), $\mathcal{A}$ actions, and $\mathcal{T}$ task specifications (e.g., language prompts). A policy $\pi_\theta(a_t \mid s_t, T)$ maps state $s_t \in \mathcal{S}$ and task $T \in \mathcal{T}$ to a distribution over actions $a_t \in \mathcal{A}$. We assume access to a pretrained generalist policy $\pi_{\theta_{\text{pre}}}$, trained on a diverse set of tasks and environments, and denote its training data as $\mathfrak{D}_{\text{pre}}$. For a new *target task* $T_\eta$ (e.g., "wipe the whiteboard"), we assume access to a demonstration dataset $\mathfrak{D}_\eta = \left\{ (s_t^{(i)}, a_t^{(i)}, T_\eta) \right\}$. In general, we assume that $\mathfrak{D}_\eta$ is collected in a single (or small number of) environment $\mathcal{M}_\eta$, and $|\mathfrak{D}_\eta| \ll |\mathfrak{D}_{\text{pre}}|$.

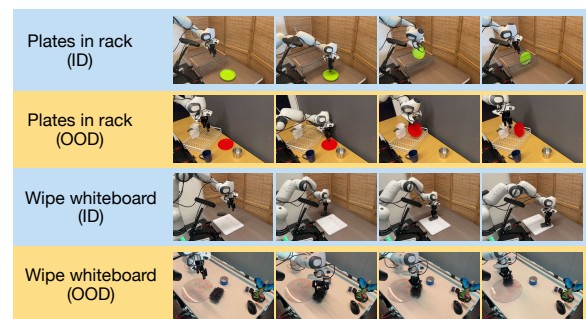

Figure 3: Example filmstrips of the in-distribution (ID) and out-of-distribution (OOD) tasks from DROID robot experiments in Section 6.

**Behavioral cloning & finetuning.**    For adapting the policy to the target task, we consider the standard behavioral cloning (BC) objective. For policy parameterization $\pi_\theta$ and demonstration dataset $\mathfrak{D}$, the training objective is:

$$\mathcal{L}_{\mathrm{BC}}(\theta\,;\mathfrak{D}) \;:=\; -\frac{1}{|\mathfrak{D}|} \sum_{(s_t,a_t,T)\in\mathfrak{D}} \log \pi_\theta(a_t \mid s_t, T). \tag{1}$$

We consider two main finetuning settings: **Task-finetuning (task-FT)**, in which we train exclusively on the target dataset $\mathfrak{D}_\eta$ (e.g., because pretraining data is proprietary); and **co-finetuning (co-FT)**, where we finetune on a mix of $\mathfrak{D}_\eta$ and $\mathfrak{D}_{\mathrm{pre}}$ to help preserve pretraining capabilities (e.g., in case of open-source pretraining datasets).

**Evaluation.**    In practice, when we finetune a policy, we don't simply want it to work only in the setting where we collected the finetuning demonstration, but for it to complete the demonstrated task in a variety of contexts or scenes. Current methods often fail in this regime because they overfit heavily to the small finetuning dataset. Therefore, to assess overfitting and robustness of the finetuned policy, we evaluate the performance of finetuned policies in the following three settings:

1. **Target task in-distribution (ID)**: measures policy performance on the target task $T_\eta$, with objects, initial poses/layouts, camera placements, lighting conditions, and backgrounds *observed* in the finetuning dataset $\mathfrak{D}_\eta$.

2. **Target task out-of-distribution (OOD)**: measures the performance on the target task, in scenarios *not* observed in the finetuning dataset, such as changes in object instances, backgrounds, lighting conditions and camera angles. This measures the *robustness* of the finetuned policy.

3. **Generalist tasks**: measures policy performance on tasks *other than the target task*, but for which we would expect the generalist policy $\pi_{\theta_{\mathrm{pre}}}$ to perform reasonably. This measures how well the finetuned policy retains *generalist* capabilities from the pretrained model.

## 4    CHALLENGES OF FINETUNING GENERALIST ROBOT POLICIES

To understand the challenges of robust finetuning of generalist robot policies, we start by evaluating the standard finetuning approach. We evaluate finetuning the full model (Task-FT) in LIBERO (Liu et al., 2024a), a robotic manipulation simulator containing 130 total tasks. Concretely, given a state-of-the-art vision-language-action policy (Black et al., 2024), pretrained on demonstration data from 117 tasks from the LIBERO-{90, goal, spatial, object} suites, we finetune it to a new LIBERO target task. We consider three target tasks from LIBERO-10: `mugs-on-plates`,

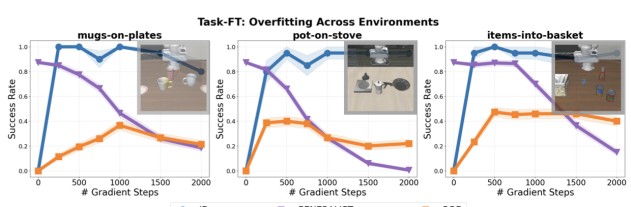

Figure 4: **The standard approach for policy finetuning often overfits.** As the policy is trained for more gradient steps, it performs worse on tasks other than the new target task ("**GENERALIST**") and may even start to degrade on scenarios seen in the finetuning data ("**ID**"). Most importantly, it is not able to transfer the generality of a base policy to do well under variations of the target task (new object positions, instances, viewpoints; "**OOD**").

`pot-on-stove`, and `items-into-basket`. We measure performance in the three scenarios introduced in Section 3: ID, OOD, and Generalist. For OOD evaluations, we alter object positions, add new distractors, and change backgrounds. Generalist evaluations are performed over 20 tasks from the pretraining dataset. More details on OOD evaluations in Section 6.1 and Section A.7.

Fig. 4 shows the performance of Task-FT on three types of evaluations. We find a clear tradeoff between generalist and ID target task performance: though the model gets better in ID target task after fine-tuning, it increasingly loses its generalist capabilities. Additionally, when the model is finetuned for too long, it even starts losing performance for ID tasks. Both of these phenomena are likely because the model has severely overfitted to the small demonstration dataset, and is unable to do other tasks or recover from small mistakes unseen in the dataset. More importantly, one would hope that the finetuned model can generalize to small variations of the target task not seen in the

finetuning dataset, since the pretrained model contains knowledge for such generalization. However, all tested checkpoints above show a large gap between ID and OOD performance, showing that the model is not able to complete the target task when there are small variations. Section A.4 shows that even with careful tuning of learning rate and number of gradient steps, such overfitting still exists.

## 5  RETAIN: ROBUST POLICY FINETUNING VIA MODEL MERGING

The previous section illustrates that standard finetuning approaches cause the model to quickly forget generalist capabilities, and fail to transfer the pretrained policy's robustness to the target task. To address these issues, we propose RETAIN (**R**obust fin**E**-tuning wi**T**h p**A**rameter merg**IN**g), a simple approach for robust finetuning of robot policies. Given pretrained policy weights $\theta_{\text{pre}}$ and finetuned policy weights $\theta_{\text{ft}}$, we propose linearly interpolating $\theta_{\text{pre}}$ and $\theta_{\text{ft}}$ to obtain a final policy checkpoint, $\tilde{\theta}$. So, RETAIN produces a final policy $\pi_{\tilde{\theta}}$ by setting:

$$\tilde{\theta} = (1 - \alpha) \cdot \theta_{\text{pre}} + \alpha \cdot \theta_{\text{ft}} \tag{2}$$

for $\alpha$, a tunable merging weight. Though surprisingly simple, as we will see, this weight space "merging" of pretrained and finetuned checkpoints leads to significantly improved OOD performance on the target task, while retaining generalist policy capabilities (see Section 6.5). While weight merging itself already improves the policy's ability to retain and pass on generalist abilities, in the following we introduce two further improvements: utilizing the pretraining data $\mathfrak{D}_{\text{pre}}$ (in settings where it is available) to augment our task data $\mathfrak{D}_{\eta}$ during finetuning, and merging $\theta_{\text{pre}}$ and $\theta_{\text{ft}}$ in a modality-specific manner. We introduce these two methods below, and show how RETAIN can also enable *continual* adaptation to new tasks.

### 5.1  CO-FINETUNING

In Eq. (2), the finetuned policy weight $\theta_{\text{ft}}$ can either be optimized via task-finetuning or co-finetuning, as described in Section 3. In situations where the pretraining dataset, or a subset of it, is available, we can finetune the policy weight on a mix of $\mathfrak{D}_{\text{pre}}$ and $\mathfrak{D}_{\eta}$. Such co-finetuning usually leads to better retention of generalist abilities after finetuning (Bousmalis et al., 2023; Fu et al., 2024b; Dass et al., 2025). We find that we can use model merging together with co-finetuning, which we will refer to as RETAIN-co-FT, to enable greater generalization on the target task and better preserve generalist knowledge than model merging with task fine-tuning, which we call RETAIN-task-FT (see Section 6).

### 5.2  MODALITY-SPECIFIC MERGING

While prior works have explored model-merging in the context of uni-modal vision or language models (Lu et al., 2025; Jang et al., 2024; Matena & Raffel, 2022; Yang et al., 2023a; Jin et al., 2022; Wang et al., 2024a), robotics is fundamentally a multi-modal problem. Modern generalist robot policies are typically instantiated as vision-language-action (VLA) models (see Fig. 2) that consist of a vision encoder ($v$), a large language model backbone ($l$), and often an "action expert" module that decodes robot action outputs ($a$). We find that, in such multi-modal settings, it can be advantageous to use separate merging weights for different modalities. As such, we can expand the RETAIN merging objective to:

$$\begin{pmatrix} \tilde{\theta}_v \\ \tilde{\theta}_l \\ \tilde{\theta}_a \end{pmatrix} = \left[ 1 - \begin{pmatrix} \alpha_v \\ \alpha_l \\ \alpha_a \end{pmatrix} \right] \cdot \begin{pmatrix} \theta_{\text{pre},v} \\ \theta_{\text{pre},l} \\ \theta_{\text{pre},a} \end{pmatrix} + \begin{pmatrix} \alpha_v \\ \alpha_l \\ \alpha_a \end{pmatrix} \cdot \begin{pmatrix} \theta_{\text{ft},v} \\ \theta_{\text{ft},l} \\ \theta_{\text{pre},a} \end{pmatrix} \tag{3}$$

Section 6.4 shows that, surprisingly, it often suffices to only merge the language model parameters.

### 5.3  CONTINUAL TASK ADAPTATION

We observe that RETAIN enables finetuned policies to retain the generalist capabilities of the pretrained policy. As such, we can use RETAIN to *sequentially* add tasks into a pretrained checkpoint by iteratively merging finetuned weights into the base model and continuing to finetune from the merged checkpoint (see

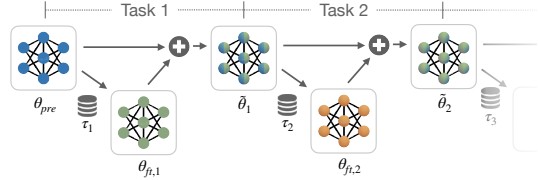

Figure 5: **RETAIN enables continual merging of new skills into generalist policy backbones.**

Fig. 5). Formally, for a *sequence* of target tasks $T_{\eta_1}, \ldots T_{\eta_N}$ we can compute a sequence of adapted RETAIN policies that accumulate new task capabilities as:

$$\tilde{\theta}_n = (1 - \alpha) \cdot \tilde{\theta}_{n-1} + \alpha \cdot \theta_{\text{ft,n}} \big|_{n \in \{1\ldots N\}} \tag{4}$$

where $\theta_{\text{ft,n}}$ denotes the parameters finetuned on the $n$th task.

## 6 EXPERIMENTS

The goal of our experiments is to evaluate RETAIN's ability to *robustly* finetune generalist policies to new tasks, i.e., to broaden the finetuned policy's ability to generalize to unseen settings in the target task. Concretely, we aim to answer the following questions: **(1)** Can RETAIN learn a new skill robustly and generalize more broadly to variations of the skill than prior finetuning approaches? **(2)** What factors influence whether we can effectively merge pretrained and finetuned policy? **(3)** Can RETAIN enable continual merging of a sequence of several skills into the pretrained policies?

### 6.1 EXPERIMENTAL SETUP

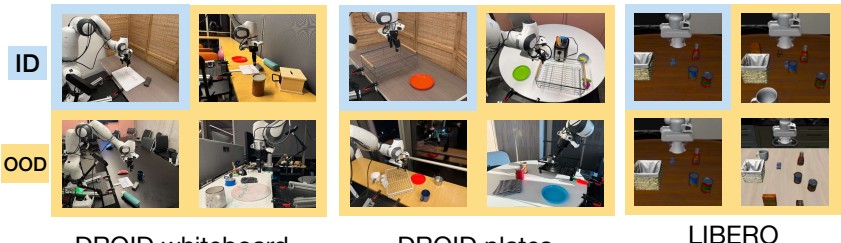

DROID whiteboard     DROID plates     LIBERO items-into-basket

Figure 6: We evaluate policy finetuning on two real-world DROID finetuning tasks (**left**, **middle**) and three simulated LIBERO finetuning tasks (**right**, only one visualized here). In each task, we collect a modest number of demonstrations (50–100) in a comparatively narrow setting (**blue**), but evaluate on a much broader set of variations for the same task (**yellow**), including variations to scene, object instances, initial positions, lighting conditions, distractors, and viewpoints. This tests transfer of the generalization ability of the pretrained policy to the target task. Example trajectories in Fig. 3.

**Environments and tasks.** We evaluate RETAIN in real-world and simulated finetuning settings (see Fig. 6). For our **real-world experiments**, we use the DROID robot setup (Khazatsky et al., 2024), which consists of a 7-DoF Franka robot arm with a wrist-mounted camera and at least one external camera. We design two challenging fine-tuning tasks: wiping the whiteboard with an eraser (which we call `whiteboard`) and putting the dishes into a drying rack (which we call `plates`). For both tasks, we collect roughly $50$ and $100$ demonstrations, respectively. All demonstrations are collected in a single environment with minor position variations (see Fig. 6, blue), to mirror the variation in typical narrow-data finetuning regimes (Zhang et al., 2025). We test generalization of the finetuned policies on a much broader set of target environments (Fig. 6, yellow), which include unseen backgrounds, object instances, and camera views. See example task trajectories in Fig. 3. For our **simulated experiments**, we use the LIBERO simulation environment (Liu et al., 2024a). We finetune policies pretrained on the LIBERO-{object, spatial, goal, 90} datasets to three new tasks from the LIBERO-10 suite: `pot-on-stove`, `mugs-on-plates`, and `items-into-basket`. We use $\approx 45$ demonstrations per task, obtained after filtering and preprocessing the 50 demos provided with the LIBERO simulator, which only contain minor variations to the initial positions of each object, and again test on a much broader distribution of initial positions, backgrounds, and additional distractors (see Fig. 6, right). More setup details in Section A.6 and Section A.7.

For both LIBERO and DROID, we evaluate our method and baselines on three different OOD scenes. To pick the merging coefficient $\alpha$, we use one OOD scene as the "validation" scene, and tune the hyperparameter $\alpha$ for best performance on that scene. We only use $\alpha \in \{0.25, 0.5, 0.75\}$ in DROID. Then, we use the rest of the OOD scenes as the "test" scenes without any hyperparameter tuning.

**Pretrained policies.** We use state-of-the-art pretrained robot policies for our experiments. For our real-world DROID experiments, we use $\pi_0$-FAST-DROID (Pertsch et al., 2025), the best open-source DROID policy at the time of our experiments. For LIBERO, we use a $\pi_0$ (Black et al., 2023)

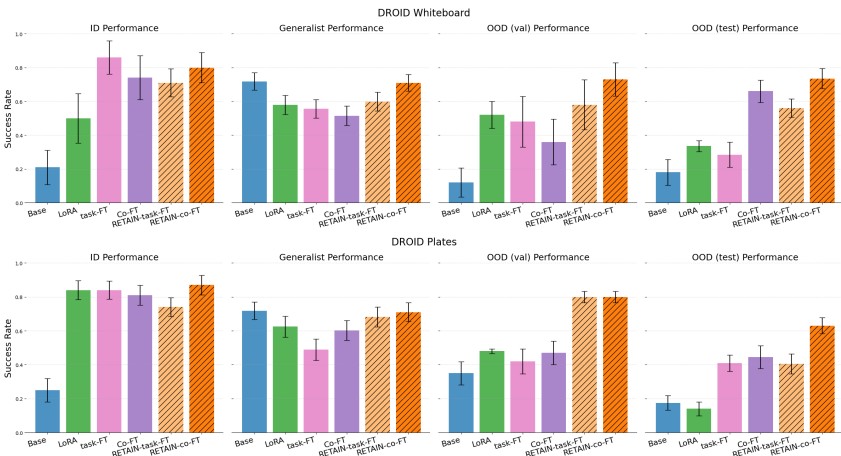

Figure 7: RETAIN results on two DROID tasks, `whiteboard` (top) and `plates` (bottom). **RE-TAIN significantly outperform baselines in OOD evaluation and is competitive in ID evaluations**, showing that it is able to learn new skills robustly and can generalize to its variations using pretrained knowledge. **RETAIN also does best on generalist evaluations**, showing that it is best at retaining abilities to solve old tasks. We tune merging coefficient $\alpha$ on one "val" OOD scene, and use the same value for two other "test" OOD scenes.

policy fine-tuned on the LIBERO-{object, spatial, goal, 90} datasets as our pretrained policy. $\pi_0$ is a generalist policy with a flow-based action expert, while $\pi_0$-FAST-DROID is an autoregressive transformer based on next-token prediction.

**Comparisons.** We compare against prior methods, including those that incorporate regularization techniques to reduce overfitting, in the robust finetuning setting. Specifically, we compare our approach, RETAIN, to: "**Task-FT**", which finetunes the pretrained policy only on the target task dataset using behavioral cloning (Eq. (1)) (Bain & Sammut, 1995; Black et al., 2024); "**Co-FT**", which finetunes on a mix of pretraining and target task data to reduce overfitting (Fu et al., 2024b; Dass et al., 2025), "**LoRA**", which uses low-rank adaptation (Hu et al., 2022) during finetuning to retain more of the pretraining capabilities (Mittenbuehler et al., 2023; Kim et al., 2024); "**Freeze-FT**", which freezes the language model backbone during finetuning and only updates the vision encoder, and, in the case of $\pi_0$, the action expert output head, following similar approaches in prior work, e.g., Kim et al. (2024); Zhang et al. (2025); **Scratch**, which learns a policy from scratch on the demonstration dataset instead of finetuning a generalist policy. Section A.9 details policy classes and implementation. Section A.8 details hyperparameters choices and tuning process.

## 6.2 RETAIN LEARN NEW SKILLS ROBUSTLY AND GENERALIZE MORE BROADLY

**RETAIN solves the finetuning task in a broader range of variations.** We compare RE-TAIN with the aforementioned baseline methods when finetuning to a new task, and show the performance on the three types of evaluations. Fig. 7 shows results on two DROID environments, and Fig. 8 shows the average of three LIBERO environments. The ID and OOD evaluations show how well a method learns to do the new skill: ID evaluations show whether the method learned to fit the demonstration dataset exactly, and OOD evaluations assess whether the method has learned to generalize to the same task exhibiting variations not seen in the finetuning dataset. The OOD (val) scenes show "the upper bound" in performance when $\alpha$ is allowed to be tuned slightly, while the OOD (test) scenes show the performance with no tuning.

In real-world **DROID environments (Fig. 7)**, all methods perform much better on ID evaluations after finetuning, showing that the policy has adapted to the new task in the exact same context as the demonstration dataset. On ID evaluations, methods that use regularization, such as LoRA and Co-FT, perform slightly worse on `whiteboard`, likely because they are too constrained to adapt well to the new task. On OOD (test) evaluations, baseline finetuning methods perform significantly worse: while they can complete the new task with $70-80\%$ success rate in the ID setting, they have $30-50\%$ success rate on average in the OOD setting. This shows that the baseline methods are very sensitive to small variations (such as object change, location change, scene change) and cannot generalize

to perform well. In comparison, both RETAIN-task-FT and RETAIN-co-FT perform much better, achieving more than $60\%$ on `plates` and close to $80\%$ on `whiteboard` OOD evaluation. Note that this is similar to the policy's ID evaluation performance on `whiteboard`, suggesting that RETAIN can perform a *generalized* new skill with the same performance regardless of variations. Comparing the performance of OOD val and test sets, we see that the performance is sometimes not impacted (`whiteboard`) and sometimes better with slight hyperparameter tuning (`plates`). This shows that RETAIN is somewhat robust to hyperparameter, but can get even better when tuned for the particular OOD scene. Overall, in OOD evaluations, RETAIN enables the policy to outperform both the base model and other finetuned models.

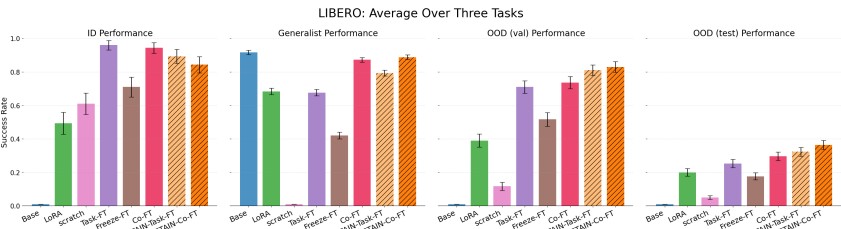

Figure 8: RETAIN results averaged over the three LIBERO tasks. Similar trend as Fig. 7.

In **LIBERO environments (Fig. 8)**, all methods exhibit trends similar to those in DROID environments. Since the LIBERO simulation is a much easier task than real world robotic tasks, certain baseline fine-tuning methods achieve near-perfect performance in ID evaluations. Under OOD evaluations, both RETAIN-task-FT and RETAIN-co-FT help improve the policy's robustness to scene variations. We observe that the improvement in OOD performance over baselines such as Co-FT in LIBERO is smaller than that in DROID, and we attribute this to the lack of generalist capabilities of the base model. We offer more discussion of this in zzz

**RETAIN still performs well on tasks from the pretraining distribution.** As shown above, RETAIN allows the model to *not* overfit to the finetuning dataset and generalize to solve a broader distribution of the finetuning task. One natural question is whether RETAIN has overfit to the finetuning task distribution, and whether it can still solve tasks under the pretraining distribution. To evaluate retention of generalist skills, we evaluate RETAIN and baselines under our generalist evaluation scenes. For DROID, we evaluate on $44$ different real-world tasks; for LIBERO, we evaluate on 20 random tasks in the LIBERO pretraining dataset (5 each from LIBERO-{object, spatial, goal, 90}). Fig. 7 and Fig. 8 (2nd subfig) show that RETAIN performs just as well as the pretrained model on generalist evaluations, showing that it has not lost its ability to solve old tasks from pretraining. RETAIN works even better when combined with co-finetuning, as discussed below.

**Model merging performs better with co-finetuning.** In Fig. 7 and Fig. 8, RETAIN-co-FT outperforms RETAIN-task-FT in all three evaluation settings in almost all tasks. RETAIN-co-FT is particularly effective at improving performance in generalist and OOD evaluation. In fact, we observe that co-FT almost always helps improve performance over task-FT in generalist evaluation, but not always in OOD evaluation. We hypothesize this is because co-finetuning and model merging play different roles in the regularization process: co-finetuning helps the finetuned model not overfit to the small target dataset by continuous training on pretraining data, but does not elicit pretrained knowledge to help generalization on the new task; on the other hand, model merging explicitly tries to elicit pretrained knowledge and combine it with finetuning knowledge in parameter space; however, doing so with a task-FT model is worse at keeping pretraining abilities because the task-FT model has overfitted to the target dataset.

## 6.3 RETAIN SCALES WITH THE AMOUNT OF PRETRAINING DATA

In this section we ask: how well does RETAIN do when the pretrained model contains differing amounts of "generalist knowledge", as induced by different amounts of pretraining data? To study how differing amounts of pretraining data affects RETAIN's performance, we use the DROID dataset, containing a large quantity of diverse robot data. We consider three pretrained generalist VLA policies: (1) the public $\pi_0$-FAST-DROID checkpoint that is trained on all of DROID and a large repertoire of robot data from Physical Intelligence Pertsch et al. (2025), (2) only trained on all of DROID, totaling $76k$ episodes, and (3) only trained on a subset of DROID of $20k$ episodes.

All three policies are pretrained by taking roughly 1 epoch over the dataset, and finetuned on the `plates` task with the same hyperparameters.

We then evaluate the three RETAIN-co-FT policies, each obtained by merging the respective pretrained policy with its finetuned counterpart, on ID and OOD scenes in Fig. 9. To account for the performance difference of the base policies, we report the success rate increase between RETAIN-co-FT and co-FT. In ID evaluations, policies trained with differing amount of data perform similarly. In OOD evaluations, more pretraining data leads to significantly better performance. This shows that RETAIN scales with the amount of pretraining data, and can better "transfer" generalist knowledge from the pretrained model to new scenes

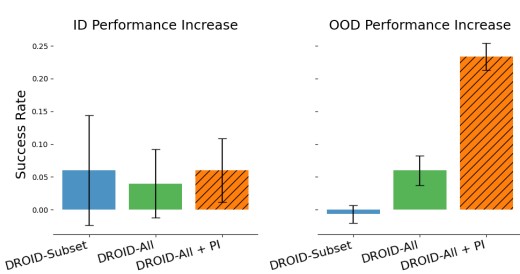

Figure 9: **RETAIN performs better on OOD tasks when the pretrained generalist policy is trained on more data**. OOD performance is averaged across three `plates` scenes.

when the pretrained model is more general. In fact, the best performing model (DROID-ALL + PI) performs nearly as good on OOD evlauation as it does on ID evaluation, as shown in Fig. 15.

## 6.4    ANALYZING THE IMPORTANCE OF MERGING DIFFERENT PARAMETERS IN RETAIN

Here we seek to build a mechanistic understanding of how the merging coefficient impacts the performance of merged models. In Fig. 10, we plot OOD performance against $\alpha$ averaged across three LIBERO tasks, each with three types of OOD scenes. $\alpha = 0$ corresponds to the pretrained model, and $\alpha = 1$ corresponds to the co-finetuned model. Model merging ($0 < \alpha < 1$) helps improve model performance in OOD evaluation, as long as the merged model is not too deviated from the finetuned model. When the merged model is too similar to the pretrained model, it does not have enough task-specific knowledge, and has near 0 performance.

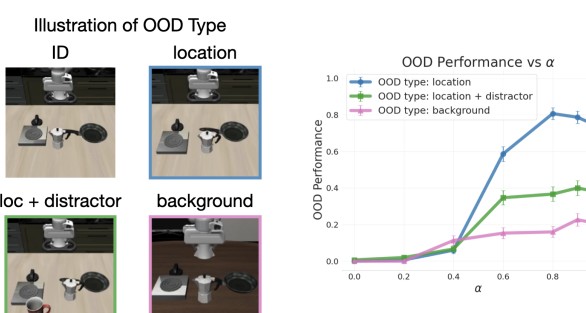

Figure 10: **Left:** Illustration of three different kinds of OOD scenes we consider in LIBERO, shown in `pot-on-stove`. **Right:** Plot shows how the OOD performance on three types of OOD variations changes with the merging parameter $\alpha$, averaged across three LIBERO envs.

And as we have shown in Fig. 8, the model with the $\alpha$ value that has the best OOD performance is also comparable to baselines in ID evaluations and better at generalist evaluations.

Next, we explore whether merging different parameters with different coefficients has an impact on the merged policy's performance. Specifically, we consider modality-specific merging in the context of VLA policies. As explained in Section 5.2, we use separate coefficients for merging the parameters of the vision encoder, language model, and action expert. Fig. 11 (left) shows the OOD performance of the merged model as we vary $\alpha_v$, $\alpha_l$, and $\alpha_a$ from 0 to 1 on the `mugs-on-plates` task: dark colors represent low OOD performance, and light colors represent high OOD performance. Observe that the cube has the largest color gradient in the $\alpha_l$ direction: this shows that the language model parameters have the most influence on performance. The best performing models has $\alpha_l = 0.8$ (see the highlighted plane at $\alpha_l = 0.8$ in Fig. 11 (left) for the brightest colored dots). Next, to understand how $\alpha_v$ and $\alpha_a$ impact performance, we plot in Fig. 11 (middle) the change in OOD performance with these two coefficients when averaged over $\alpha_l$. This 2D plot essentially squashes the 3D cube plot in the language direction. Unlike the behavior of $\alpha_l$, higher values of $\alpha_a$ and $\alpha_v$ lead to better performance; the best performance is achieved at $\alpha_a = \alpha_v = 1$.

These results suggest that during model merging, it may suffice to only merge the parameters of the language model backbone ($\alpha_l < 1$) and leave the parameters in the vision encoder and the action expert set to the parameter values in the finetuned model ($\alpha_a = \alpha_v = 1$). To validate this hypothesis, we compare the OOD performance of merging all parameters with RETAIN ($0 < \alpha < 1$) to only

merging language model parameters. In Fig. 11 (right), we plot the performance of the two merging schemes over three different LIBERO tasks, each averaged over three types of OOD scenes. This shows that the two merging schemes achieve very similar performance, indicating that we only need to merge parameters from the language model backbone (instead of all parameters) to inherit the model's generalization ability and robustness to variations in the target scene.

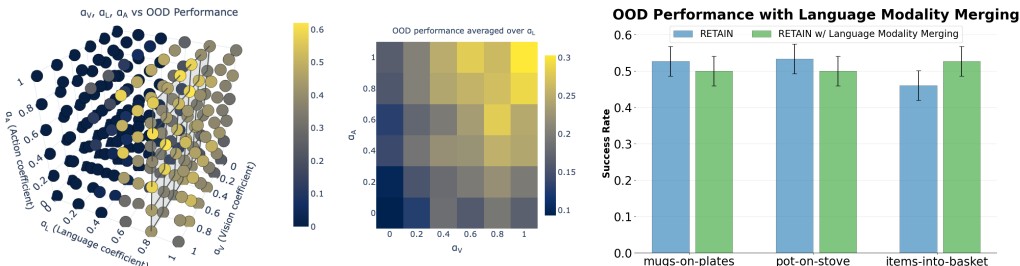

Figure 11: **Language model parameters have the most influence in modality-specific merging.** **Left**: Merged model's OOD performance over a grid search of $\alpha_a, \alpha_v, \alpha_l$, and $\alpha_l$ has the most impact. **Middle**: OOD performance of $\alpha_a$ and $\alpha_v$ averaged over different $\alpha_l$, and higher values are better. **Right**: Merging only the language model parameters ($\alpha_a = \alpha_v = 1, \alpha_l < 1$) performs similarly to merging all parameters.

## 6.5 RETAIN ENABLES ROBUST LEARNING OF MULTIPLE SKILLS SEQUENTIALLY

Finally, we test whether RETAIN can enable learning multiple skills in sequence, as described in Section 5.3, and still retain its generalist abilities. We consider learning the two DROID tasks sequentially, first finetuning on `plates`, and then using this as an initialization to finetune on `whiteboard`. As outlined in Section 5.3, RETAIN uses the merged model from the first stage of finetuning as initialization for the second finetuning stage. During evaluation time, we test whether the final policy, after it has sequentially been trained on both tasks, can solve both tasks under ID and OOD evaluations. We compare against co-FT in the sequential learning setting, since it is the strongest baseline at retaining prior knowledge in single-task finetuning (see Fig. 7). Fig. 12

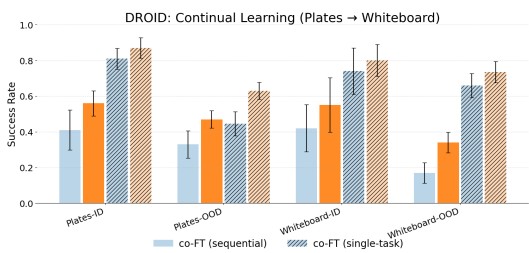

Figure 12: **RETAIN enables continual adaptation to a sequence of two skills.** Evaluation results show the performance of the final policy after sequentially finetuning on two tasks, evaluated on different scenes. OOD performance averaged across two test scenes.

shows the performance of the two policies on the two tasks under both ID and OOD settings. We also plot the performance of these two methods in the single-task finetuning setting in dashed lines. These two comparisons serve as the oracle performance ceiling that we expect on these tasks, and is not meant as baselines for the sequential setting. When evaluated on the first task, `plates`, RETAIN does much better than co-FT in the sequential setting, showing that it is better at retaining its ability to solve the first task even after a second round of finetuning. When evaluated on the second task, `whiteboard`, RETAIN is also better than co-FT under ID evaluations. RETAIN outperforms co-FT in the sequential setting under all tasks and evaluation types.

## 7 CONCLUSION

We present a simple yet effective method, RETAIN, for robust finetuning of generalist robot policies. We show that by simply interpolating the weights of a generalist policy before and after it is finetuned on a target task, we can "merge" the generalization ability of the base policy with the task-expertise of the finetuned policy. Through comprehensive real world and simulated experiments, we show that RETAIN can help the policy generalize significantly better to variations of the target task unseen in the demonstration, and is able to retain performance on general tasks. We also apply RETAIN to sequentially acquire new skills in a lifelong learning setting, and find that it can robustly "merge" skills into a single policy.

## 8 LIMITATIONS

While we empirically verified that RETAIN works exceptionally well in helping the finetuned model generalize to out-of-distribution variations of the task, we don't understand the full scope of the reasons why model parameter merging was able to lead to such generalization. This is an interesting area for future work. We have included some discussion in Section A.10 of some hypothesess and why previous work found model parameter merging effective for vision and language tasks. Additionally, RETAIN involves an important hyperparameter, the merginge coefficient, that can be tuned. While we find in our real world experiments that RETAIN is robust to different values of this parameter, slight tuning of this parameter is needed. One avenue of future work is determining a good heuristic of how to choose this value.

## 9 ACKNOWLEDGEMENTS

This research was partly supported by ONR N00014-25-1-2060. This research used the Savio computational cluster resource provided by the Berkeley Research Computing program at UC Berkeley. We would also like to thank the anonymous ICLR reviewers for thoughtful comments and suggestions.

## 10 REPRODUCIBILITY STATEMENT

We describe all the implementation details in Section A.6 and Section A.7, and hyperparameter choices in Section A.8, which should enable researchers to reproduce our results. We also remark that our algorithm is extremely simple to implement. We will share the code during the review process and also release the code publicly.

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

## A  APPENDIX

### A.1  DROID RESULTS DETAILS

Fig. 7 reports performance of two DROID tasks: `whiteboard` and `plates`. Here we present the average of the two tasks. The OOD performance shown here is averaged across the val and test scenes.

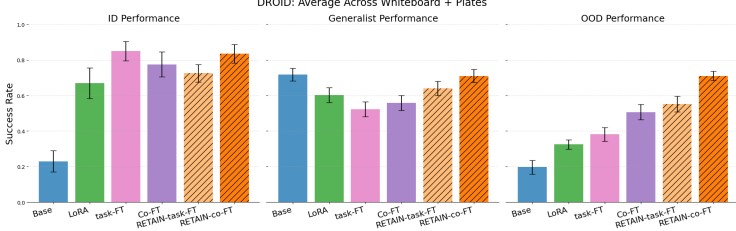

Figure 13: Average results on performance of the two DROID tasks: `whiteboard` and `plates`.

### A.2  LIBERO RESULTS DETAILS

Fig. 8 reports the average performance of three LIBERO tasks: `pot-on-stove`, `mugs-on-plates`, and `items-into-box`. Here we present the individual performance for all three tasks in Fig. 14. The OOD performance shown here are averaged only across the two test scene.

Additionally, we offer here more discussion on why we see less improvement of RETAIN over baselines in LIBERO as compared to DROID. For DROID, the base model $\pi_0$-FAST was trained on $76k$ diverse trajectories in $564$ scenes, while the LIBERO base model is only trained on $5.3k$ trajectories in 117 scenes with fairly limited diversity. As such, the LIBERO base model contains much less generalist ability than the DROID base model. And as we will shown in Section 6.3, merging with a less general base model inherits less generalization power, giving less improvement under OOD evaluations. There is also a much bigger gap between val and test OOD scenes, because the different types of variations for each OOD scene has a different difficulty level for the limited generalist model in LIBERO.

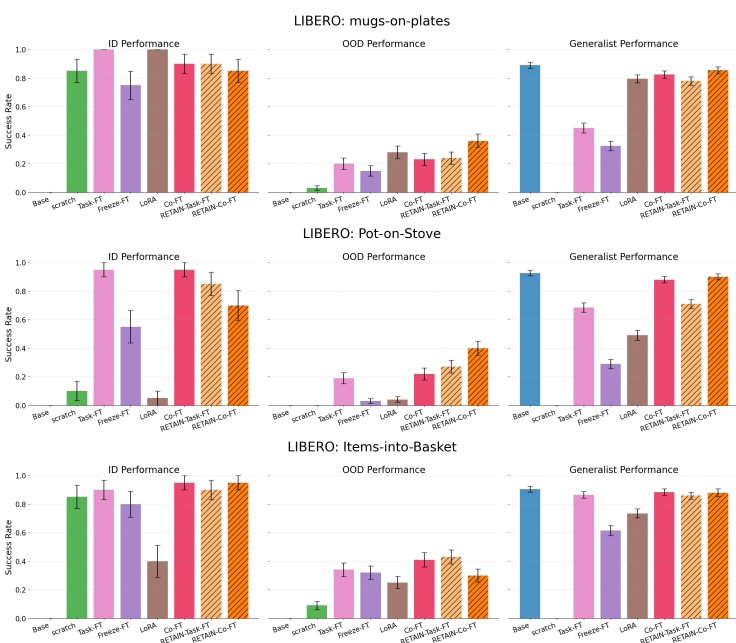

Figure 14: Detailed results on performance of the three LIBERO tasks: `pot-on-stove`, `mugs-on-plates`, and `items-into-box.`

## A.3 DATA SCALING RESULTS DETAILS

Fig. 15 highlights more details of the data scaling results, examining for which base models RETAIN provides the greatest benefit: it shows the ID/OOD performance for co-FT and RETAIN-co-FT with varying amounts of pretraining data that led to the difference results presented in Fig. 9. We see that as we scale pretraining data, ID performance is similar between co-FT and RETAIN-co-FT, while the benefit provided by RETAIN on OOD greatly increase with the generalist capabilities of the base model, and scales we pre-training data.

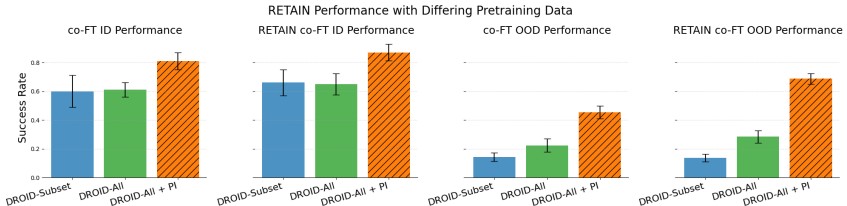

Figure 15: Detailed results on the benefits of RETAIN with increasing pretraining data.

## A.4 ABLATION ON LEARNING RATE AND GRADIENT STEPS

Here, we ablate the learning rate and number of gradient steps we take in the task-FT policy in Fig. 4 to study whether better hyperparameter choices can reduce or resolve overfitting[0]. In Fig. 4, we use learning rate $2.5e - 5$. Here in Fig. 16, we ablate four different learning rates: one greater than the original and two smaller. We evaluate the model at every 100 gradient steps to also ablate on the number of gradient steps we take. With a larger learning rate, it's clear that the overfitting issue is more severe, and the performance on all three kinds of evaluations (ID, OOD, Generalist) go down to

---

[0]Following the best practice from Black et al. (2024), we always use a learning rate warmup period of 1000 steps in LIBERO and a consine decay schedule over $30k$ steps to $1/10$ of the peak learning rate. The learning rate we report here is the peak learning rate.

near zero. With a smaller learning rate, the model still performs well in ID evaluations, and suffers less from forgetting generalist capabilities (measured from Generalist evaluations). This is to be expected because the finetuned model is closer to the pretrained model with a smaller learning rate. However, note that with a smaller learning rate, the OOD evaluation performance is also worse. We plot the OOD performance achieved using the original learning rate in Fig. 4 as the dotted orange line in Fig. 16, and the gap between the dotted and solid orange line shows the performance gap in OOD evaluation when we lower the learning rate, possibly due to underfitting. This experiment shows that tuning the learning rate and the number of update steps does not solve the problem of overfitting during finetuning. While lowering the learning rate can retain more generalist knowledge, it does not prevent the gradual loss of it. More importantly, the lower learning rate leads to worse OOD evaluation performance.

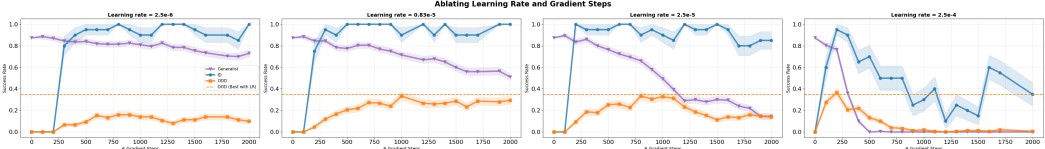

Figure 16: Ablation on learning rate and number of gradient steps for task-FT on `mugs-on-plates` task in LIBERO.

## A.5 ANALYSIS OF FINETUNING PATH IN PARAMETER SPACE

To understand why model parameter merging helps, we first try to understand here how the parameters change during fine-tuning. In particular, we are interested in understanding how linear the finetuning path is in parameter space, since model parameter merging only moves weights in linear paths.

To start out, we check the colinearity of the vectors $\theta_{i+1} - \theta_i$ and $\theta_i - \theta_{i-1}$, where $\theta_i$ is the parameter of the finetuned checkpoint at gradient steps $100 * i$ (i.e. we plot the difference vector every $100$ steps during finetuning). We measure the colinearity as the cosine similarity between the two difference vectors. Fig. 17 shows this for the four different learning rates we ablated in Fig. 16. Since no values are close to $1$, this shows that the changes in parameter space is highly non-linear. This is expected since the parameter trajectory of deep neural networks is often highly non-linear.

Next, we attempt to more directly visualize the path/direction of the parameters during finetuning by projecting them down to 2D with Principal Component Analysis (PCA). Specifically, we again consider the difference of the weight vectors, $X_i = \theta_{i+1} - \theta_i$, at every $100$ steps during finetuning. In Fig. 18, we plot the first two principal components of $X_i$ in blue, and label the points $i$. Indeed, we see that for all learning rates, the direction of the parameters is highly non-linear. With small learning rates, the direction oscillates a lot; with larger learning rates, the direction bends in a certain direction. In addition, we plot the two principal components of the parameter-merged model as well in orange. As expected, since the the merged model takes a linear path and the finetuned one does

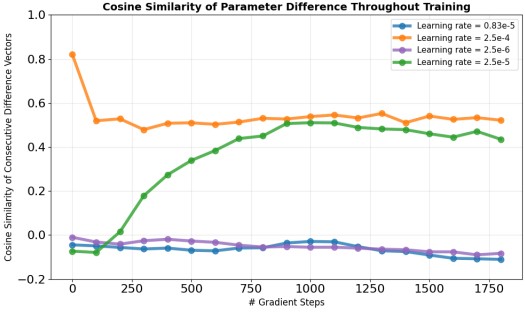

Figure 17: Cosine similarity of parameter difference vectors during finetuning, showing that the finetuning path is highly non-linear.

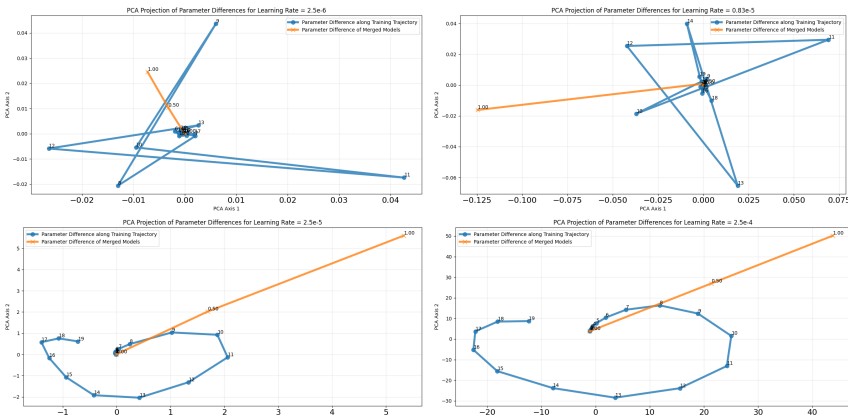

Figure 18: PCA projection of the parameter difference during finetuning to 2D. Each subplot corresponds to a different learning rate.

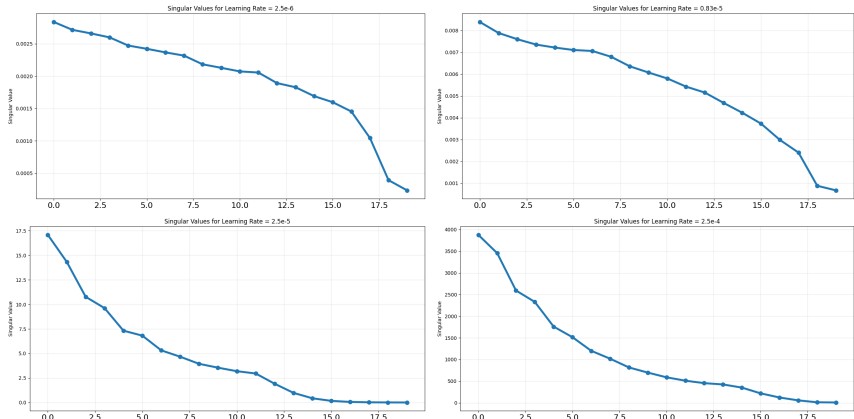

Figure 19: Singular Values of the parameter differences during finetuning. Each subplot corresponds to a different learning rate.

not, the two models end in in very different places in the parameter space. This shows that model merging achieves a different solution than any checkpoints on the finetuning path.

Finally, we analyze changes in all directions of the difference vectors $X_i$, instead of just the two principal components as shown in Fig. 18. We take the difference vector matrix $Y = [X_1; X_2; ..., X_n]$ and compute the singular values of $YY^T$ and Fig. 19. If the parameter vectors lie in a linear path, then all difference vectors would point in the same direction and there should only be 1 non-zero singular value. However, it's clear from Fig. 19 that most singular values are non-zero, showing that the path is non-linear in many dimensions. This generalizes the intuition from Fig. 18 to more dimensions, and shows that model merging indeed achieves a different solution than the finetuning path.

All these analysis experiments go to show that the finetuning path is highly non-linear, and therefore model parameter merging actually results in a different solution than the finetuned models.

### A.6 DROID SETUP DETAILS

This section outlines the details of the DROID setup that we used for our real-world experiments.

#### A.6.1 DATASETS: WHITEBOARD

The whiteboard task-dataset consists of 50 human tele-operated trajectories collected using a Oculus Quest 2 controller. As explained in 6.1, we collect the demonstrations with a fixed setup, with the only diversity being 5 different eraser initial positions and 2 orientations (vertical / horizontal). All demonstrations use the language prompt "wipe the whiteboard". Each step in our dataset consists of a base camera image (always collected from the right external camera), a wrist camera image, 8D-state, 8D-action, and the language instruction.

#### A.6.2 DATASETS: PLATES

The plates task-dataset consists of 100 human tele-operated trajectories collected in a similar manner as above. Again, the demonstractions are collected with a fixed setup, with the only diversity being 5 plate colors, 2 dish racks, and 2 dish rack orientations (vertical / horizontal). 80 of these 100 demos also contain distractor objects chosen from a training set of distractors. All demonstrations use the language prompt "put the plate in the dish rack".

#### A.6.3 TRAINING DETAILS

We utilize [7x joint angles, 1x gripper position] as our proprioceptive state and [7x joint velocity, 1x gripper position] as our actions both during training and inference. We also use the norm-stats of the DROID dataset, publicly available here, to normalize states and actions during training and inference by applying quantile-normalization. The base and wrist camera images go through several transforms (random crop, resizing to 224x224, and color jitter) during training.

The finetuning is performed with an action horizon of 10 environment steps, thus the policy learns to output action chunks of shape (10,8).

#### A.6.4 EVALUATION DETAILS

We use the 7-DoF Franka robot arm for our experiments. We use the same language instruction as training during evaluation, resize our images to 224x224, use the same state/action space specification, and normalize in the same manner as training.

During evaluations, we additionally binarize the policy's gripper action to 0/1 (open/close), as well as clip action magnitudes. The policy's action horizon is 10 environment steps, and during evaluation, we set the open loop horizon to 8: so, during evaluation, we receive 10 actions from the policy, execute the first 8, and then request a new action chunk. We execute the predicted actions at a control frequency of 15 Hz.

The policy is served on a NVIDIA H200/H100 throughout our evaluations. As the openpi repository specifies, inference requires at least 8 GB of VRAM.

#### A.6.5 EVALUATION CRITERIA: WHITEBOARD

For all whiteboard evaluations, we use the criteria specified in Table 1 to assign partial success. We perform 10 trials per policy evaluation, for both the ID and OOD evals.

#### A.6.6 EVALUATION CRITERIA: PLATES

For all plates evaluations, we use the criteria specified in Table 2 to assign partial success. We perform 10 trials per policy evaluation, for both the ID and OOD evals.

#### A.6.7 GENERALIST EVALUATIONS DETAILS

As detailed in Tables 3, 4, and 5, we measure policies' generalist capabilities by evaluating them on 44 tasks, distributed throughout 9 distinct scenes and 17 different language instructions. Importantly,

| Subtask | Cumulative Score |
|---|---|
| Pick up Eraser | 0.2 |
| Approach Whiteboard | 0.4 |
| Set Eraser on Whiteboard While Still Grasping It | 0.6 |
| Performs the Wiping Motion | 0.8 |
| Erases $\geq 90\%$ of Text | 1.0 |

*Penalty:* -0.1 if any of the above is done, but the eraser got flipped in the process.

Table 1: Subtasks and cumulative score for the `whiteboard` task.

| Subtask | Cumulative Score |
|---|---|
| Picks up Plate | 0.2 |
| Moves to Dish Rack | 0.4 |
| Rotates and Aligns Over a Groove | 0.6 |
| Tries to Insert Plate Into a Groove | 0.8 |
| Successfully Inserts Plate Into a Groove | 1.0 |

*Penalty:* -0.1 if any of the above is done but with the small grooves.
*Partial:* 0.5 if it tries to do the inserting motion but the plate is misoriented.

Table 2: Subtasks and cumulative scoring for the `plates` task.

to ensure fair comparison, we ensure that the initial conditions, camera angle, lighting, and all other such factors per task are kept the same across the various policies that we evaluate.

## A.7 LIBERO SETUP DETAILS

This section outlines the details of the LIBERO setup that we used for our simulated experiments.

### A.7.1 LIBERO PRETRAINING

In order to obtain a base-model to serve as the starting point for RETAIN in LIBERO, we pretrain $\pi_0$ on a mixture of LIBERO datasets. Specifically, we use 90 tasks from LIBERO-90, 9 tasks from LIBERO-object, 9 tasks from LIBERO-spatial, and 9 tasks from LIBERO-goal, for a total of 117 tasks in our pretraining dataset. For all LIBERO datasets, pretraining and finetuning, we utilize pre-processed RLDS versions gathered from here and here. These datasets consist of LIBERO demonstrations that have been preprocessed to upscale images, filter out transitions with idle actions, and remove failure trajectories. More details, such as hyperparameter choices, for the pretraining stage itself can be found in A.8. All subsequent fine-tuning and evaluation uses the normalization stats of this pretraining dataset, applying mean/standard-deviation normalization.

### A.7.2 LIBERO FINETUNING DATASETS

As mentioned in 6.1, we finetune on 3 tasks from LIBERO-10: `pot-on-stove`, `mugs-on-plates`, and `items-into-basket`. For each dataset, we obtain pre-processed and filtered versions as described above. 20 highlights what these finetuning tasks, and thus also our ID evals, look like.

The language instructions for each libero fine-tuning dataset are:

- `pot-on-stove`: "turn on the stove and put the moka pot on it"

- `mugs-on-plates`: "put the white mug on the left plate and put the yellow and white mug on the right plate"

- `items-into-basket`: "put both the alphabet soup and the cream cheese box in the basket"

| Scene Image | Task | # Trials | Randomization | Rubric |
|---|---|---|---|---|
|  | put the spoon in the dish rack | 4 | swap spoon and carrot position, 2 evals each | 1: pick up spoon; 1.5: move spoon towards dish rack; 2: put spoon in dish rack (anywhere) |
| | put carrot in bowl | 4 | swap spoon and carrot position, 2 evals each | 1: pick up carrot; 1.5: move carrot towards bowl; 2: put carrot in bowl |
| | put plate in dish rack | 2 | randomize initial position of the plate in front of the robot | 1: pick up plate; 1.5: move plate towards the dish rack; 2: put plate into dish rack (anywhere) |
|  | wipe the table | 2 | cloth initially on the left and right side of the open area | 1: move down towards cloth; 2: perform lateral ""wiping-style"" motion |
|  | put the plate on the table | 2 | plate initially on different dish rack holders (middle and end) | 1: moves towards red plate; 1.5: picks up plate; 2: places / drops plate onto the table |
|  | clean up the table | 2 | randomize initial position of paper ball on table | 1: picks up paper ball; 1.5 moves paper ball towards brown bin; 2: puts paper ball into bin |

Table 3: DROID Generalist evaluation tasks grouped by scene (Scenes 1-4). Each task contains associated trial counts, randomization details, and evaluation rubrics.

| Scene Image | Task | # Trials | Randomization | Rubric |
|---|---|---|---|---|
|  | close the drawer | 4 | two top, two bottom drawer | 1: moves towards the drawer; 2: closes drawer |
|  | put the stapler on the notebook | 2 | put stapler in higher and lower position on the table | 1: picks up stapler; 2: puts stapler on notebook |
| | put stapler in the drawer | 4 | put stapler in higher and lower position on the table, open top and bottom drawer | 1: picks up the stapler; 2: puts stapler into the drawer |
|  | clean the whiteboard | 2 | initial eraser position on the left and right of whiteboard | 1: pick up eraser; 2: perform wiping motions on the whiteboard; 3: erase the full smiley |
|  | put the marker in the cup | 4 | swap initial position of marker and cup, two local modifications each | 1: picks up marker; 1.5: rotates arm to put marker in roughly upright position; 2: puts marker in cup |

Table 4: DROID Generalist evaluation tasks grouped by scene (Scenes 5-8). Each task contains associated trial counts, randomization details, and evaluation rubrics.

We utilize [3x end effector (EEF) Cartesian position, 3x EEF rotation (roll/pitch/yaw), 1x gripper position] as both our states and actions. Similar to the DROID finetuning dataset, each step in our datasets consists of 1 base image, 1 wrist image, 7D state, 7D action, and the language instruction. As described earlier, we use the norm-stats of our pretraining dataset for normalizing states and actions during both training and inference. The base and wrist camera images go through several transforms (random crop, resizing to 224x224, and color jitter) during training.

The finetuning is performed with an action horizon of 50 environment steps, and the actions are padded to be 32-dimensional, thus the policy learns to output action chunks of shape (50, 32).

### A.7.3 LIBERO EVALUATION DETAILS

We use the LIBERO simulator for evaluation. Each ID eval is conducted for 20 episodes. Each OOD eval is conducted with 5 seeds, 10 episodes/seed. 21, 22, and 23 show the 3 types of OOD variations we test for in each of the 3 tasks.

The Generalist evals consists of 20 tasks, 5 each from LIBERO-object, LIBERO-spatial, LIBERO-goal, and LIBERO-90, and each task is tested for 10 episodes. Table 6 highlights a few tasks per LIBERO eval suite that we use in our generalist evals.

The seed controls OOD randomization such as translation of the objects, spawning random distractors, etc. We use the same language instruction as training during evaluation, resize our images to 224x224, and use the same state/action spaces.

During the evaluations, we extract the 7D actions by taking the first 7 elements from each 32-dimensional policy prediction. We let the simulator step for 10 steps before starting execution. The open loop horizon is set to 5, and follows a similar pattern as the DROID evals.

The evaluation criteria for all libero evals are 0 for failure, 1 for success, as determined by the simulator environment.

| Scene Image | Task | # Trials | Randomization | Rubric |
|---|---|---|---|---|
|  | put the black sponge in the blue bowl | 2 | any two configurations where the black sponge is not starting in the blue bowl | 1: picks up object; 2: puts object in correct location |
| | put the red bottle in the black bowl | 2 | any two configurations where the red bottle is not starting in the black bowl | 1: picks up object; 2: puts object in correct location |
| | put the watermelon in the purple bowl | 2 | any two configurations where the watermelon is not starting in the purple bowl | 1: picks up object; 2: puts object in correct location |
| | move the watermelon from the purple bowl to the blue bowl | 2 | any two configurations where the watermelon starts in the purple bowl | 1: picks up object; 2: puts object in correct location |
| | put the tape in the purple bowl | 2 | any two configurations where the tape is not starting in the purple bowl | 1: picks up object; 2: puts object in correct location |
| | put the waterbottle on the left side of the table | 2 | waterbottle starts on two different positions on the left side of the table | 1: picks up object; 2: puts object in correct location |

Table 5: DROID Generalist evaluation tasks grouped by scene (Scene 9). Each task contains associated trial counts, randomization details, and evaluation rubrics.

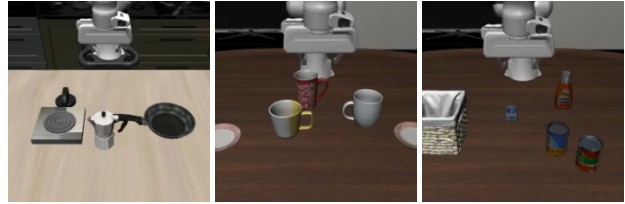

Figure 20: Three LIBERO tasks we use for finetuning: `pot-on-stove`, `mugs-on-plates`, and `items-into-basket`.

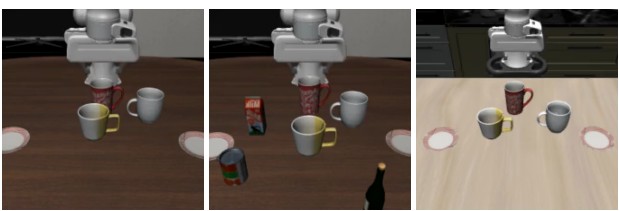

Figure 21: Three types of out-of-distribution variation of the LIBERO `mugs` task. The three different type are: (1) small translation to object positions, (2) big translation to object position and additional distractors, and (3) background change.

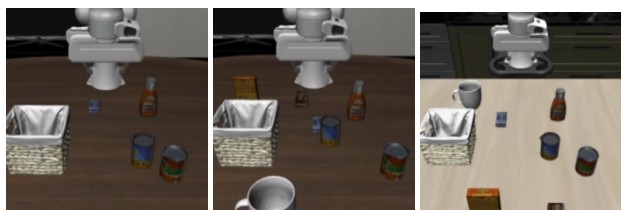

Figure 22: Three types of out-of-distribution variation of the LIBERO `items-into-basket` task. The three different type are: (1) small translation to object positions, (2) big translation to object position and additional distractors, and (3) background change and additional distractors.

| Scene Image | Task | LIBERO Eval Suite |
|---|---|---|
|  | put the bowl on the plate | LIBERO-goal |
|  | pick up the alphabet soup and place it in the basket | LIBERO-object |
|  | pick up the black bowl from table center and place it on the plate | LIBERO-spatial |
|  | put white bowl on plate | LIBERO-90 |

Table 6: Sample of LIBERO Generalist Evaluations for each evaluation suite.

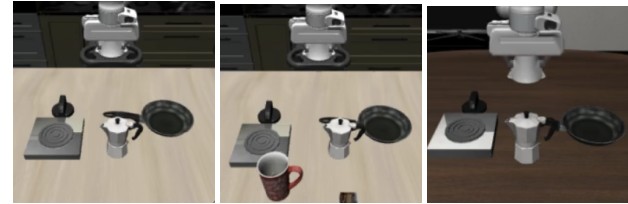

Figure 23: Three types of out-of-distribution variation of the LIBERO `pot-on-stove` task. The three different type are: (1) small translation to object positions, (2) big translation to object position and additional distractors, and (3) background change.

## A.8 HYPERPARAMETERS

### A.8.1 CHOOSING MERGING COEFFICIENT $\alpha$

For each task that we consider (e.g. `mugs-on-plate` in LIBERO, `plates` in DROID), we test policy performance on several scenes, which are different variations of the same task with different objects, distractors, and backgrounds etc. To choose what values of $\alpha$ we use, we use one OOD scene as the "validation" scene, and tune the hyperparameter $\alpha$ for best performance on that validation scene. Then, we use the rest of the OOD scenes as the "test" scenes, and report the performance of all methods only on the test scenes. In DROID experiments, we only tune $\alpha \in \{0.25, 0.5, 0.75\}$ on the validation scene, while in LIBERO experiments we tune $\alpha \in \{0.9, 0.8, 0.7, 0.6, 0.5, 0.4, 0.3, 0.2, 0.1\}$ since it is cheap to do so. We find that on DROID $\alpha = 0.5$ typically tends to perform well across tasks.

### A.8.2 CHOOSING NUMBER OF GRADIENT STEPS

For choosing how long the task-FT and co-FT should go on for, we evaluate checkpoints in ID evaluation to assess how well they fit the data. We then pick the earliest checkpoint that achieves maximal performance, to get a checkpoint that learns the target task well but does not overfit to the fientuning dataset, so that it is the strongest baseline.

### A.8.3 REAL-WORLD EXPERIMENTS' HYPERPARAMETERS

Below are tables specifying the hyperparameters we finalized upon for each of our finetuning runs for real-world experiments.

| Hyperparameter | Value |
|---|---|
| Batch Size | 32 |
| Learning Rate Schedule | Linear Warmup with Cosine Decay |
| Peak-LR | 3e-5 |
| End-LR | 2e-6 |
| Warmup-Steps | 100 |
| Decay Steps | 1000 |
| Gradient Steps | 500 |
| Weight Decay | 1e-10 |
| Optimizer | Adam(b1=0.9, b2=0.95, eps=1e-8) |
| Clip Gradient Norm | 1.0 |

Table 7: Training hyperparameters for task-FT on DROID `whiteboard`.

| Hyperparameter | Value |
| --- | --- |
| Batch Size | 32 |
| Learning Rate Schedule | Linear Warmup with Cosine Decay |
| Peak-LR | 3e-5 |
| End-LR | 2e-6 |
| Warmup-Steps | 1000 |
| Decay Steps | 10000 |
| Gradient Steps | 9999 |
| Weight Decay | 1e-10 |
| Optimizer | Adam(b1=0.9, b2=0.95, eps=1e-8) |
| Clip Gradient Norm | 1.0 |
| Cotraining-Mix | 80% task, 20% pretrain |

Table 8: Training hyperparameters for co-FT on DROID `whiteboard`.

| Hyperparameter | Value |
| --- | --- |
| Batch Size | 32 |
| Learning Rate Schedule | Linear Warmup with Cosine Decay |
| Peak-LR | 3e-5 |
| End-LR | 2e-6 |
| Warmup-Steps | 500 |
| Decay Steps | 5000 |
| Gradient Steps | 1500 |
| Weight Decay | 1e-10 |
| Optimizer | Adam(b1=0.9, b2=0.95, eps=1e-8) |
| Clip Gradient Norm | 1.0 |

Table 9: Training hyperparameters for task-FT on DROID `plates`.

| Hyperparameter | Value |
| --- | --- |
| Batch Size | 32 |
| Learning Rate Schedule | Linear Warmup with Cosine Decay |
| Peak-LR | 3e-5 |
| End-LR | 2e-6 |
| Warmup-Steps | 1000 |
| Decay Steps | 10000 |
| Gradient Steps | 5000 |
| Weight Decay | 1e-10 |
| Optimizer | Adam(b1=0.9, b2=0.95, eps=1e-8) |
| Clip Gradient Norm | 1.0 |
| Cotraining-Mix | 80% task, 20% pretrain |

Table 10: Training hyperparameters for co-FT on DROID `plates`.

| Hyperparameter | Value |
| --- | --- |
| Merging Weight for task-FT | 75% task-FT, 25% base model |
| Merging Weight for co-FT | 50% task-FT, 50% base model |

Table 11: Merging hyperparameters for DROID `whiteboard`.

| Hyperparameter | Value |
| --- | --- |
| Merging Weight for task-FT | 50% task-FT, 50% base model |
| Merging Weight for co-FT | 50% task-FT, 50% base model |

Table 12: Merging hyperparameters for DROID `plates`.

Finally, in our continual learning experiment, we use exactly the same hyperparameters as those used for plates-co-FT found in 10, just applied sequentially twice to first cotraining on the plates dataset, then on the whiteboard. In the continual learning setup, the Merging weight is also always fixed at 50% task-FT, 50% base model.

### A.8.4    SIMULATION EXPERIMENTS' HYPERPARAMETERS

In order to perform our simulation experiments, we had to first perform a round of pretraining on 117 LIBERO tasks, as described earlier. Here are the hyperparamters for this pretraining. We back-tested various checkpoints of this pretraining on the entire libero suite, and settled on step 10,000 as being a good candidate to serve as a base model, as it performed the best on both seen and unseen libero tasks.

| Hyperparameter | Value |
| --- | --- |
| Batch Size | 64 |
| Learning Rate Schedule | Linear Warmup with Cosine Decay |
| Peak-LR | 2.5e-5 |
| End-LR | 2.5e-6 |
| Warmup-Steps | 1000 |
| Decay Steps | 30000 |
| Gradient Steps | 10000 |
| Weight Decay | 1e-10 |
| Optimizer | Adam(b1=0.9, b2=0.95, eps=1e-8) |
| Clip Gradient Norm | 1.0 |

Table 13: Hyperparameters for LIBERO pretraining.

Both task-FT and co-FT on all 3 libero finetunting tasks use the same set of hyperparameters, provided below. We determine the number of gradient steps to choose by sweeping over all taken checkpoints, evaluating them on ID and OOD, and picking the best performing ones.

| Hyperparameter | Value |
|---|---|
| Batch Size | 64 |
| Learning Rate Schedule | Linear Warmup with Cosine Decay |
| Peak-LR | 2.5e-5 |
| End-LR | 2.5e-6 |
| Warmup-Steps | 1000 |
| Decay Steps | 30000 |
| Gradient Steps | 500 (items-into-basket, pot-on-stove), 1000 (mugs) |
| Weight Decay | 1e-10 |
| Optimizer | Adam(b1=0.9, b2=0.95, eps=1e-8) |
| Clip Gradient Norm | 1.0 |

Table 14: Training hyperparameters for task-FT on LIBERO-`items-into-basket`, `mugs`, `pot-on-stove`.

| Hyperparameter | Value |
|---|---|
| Batch Size | 64 |
| Learning Rate Schedule | Linear Warmup with Cosine Decay |
| Peak-LR | 2.5e-5 |
| End-LR | 2.5e-6 |
| Warmup-Steps | 1000 |
| Decay Steps | 30000 |
| Gradient Steps | 1000 (items-into-basket, pot-on-stove, mugs) |
| Weight Decay | 1e-10 |
| Optimizer | Adam(b1=0.9, b2=0.95, eps=1e-8) |
| Clip Gradient Norm | 1.0 |
| Cotraining-Mix | 50% task, 50% pretrain |

Table 15: Training hyperparameters for co-FT on LIBERO-`items-into-basket`, `mugs`, `pot-on-stove`.

As explained earlier and similar to our DROID procedure, after checking all checkpoints and picking the best-performing one, we then apply RETAIN on it to enhance its performance on OOD and Generalist evals. In simulation, we check various merging coefficients in the range [0.0, 1.0], and after doing so, here are our final merging parameters.

| Hyperparameter | Value |
|---|---|
| Merging Weight for task-FT | 90% task-FT, 10% base model |
| Merging Weight for co-FT | 90% task-FT, 10% base model |

Table 16: Merging hyperparameters for LIBERO `items-into-basket`.

| Hyperparameter | Value |
|---|---|
| Merging Weight for task-FT | 80% task-FT, 20% base model |
| Merging Weight for co-FT | 90% task-FT, 10% base model |

Table 17: Merging hyperparameters for LIBERO `mugs`.

| Hyperparameter | Value |
|---|---|
| Merging Weight for task-FT | 90% task-FT, 10% base model |
| Merging Weight for co-FT | 70% task-FT, 30% base model |

Table 18: Merging hyperparameters for LIBERO `pot-on-stove`.

### A.9 Details on Baseline Methods

Here we provide addtional details on the baseline methods we compare against.

**Task-FT** : We fine-tune all parameters of the base policy according to the behavioral cloning loss in Eq. (1). All data are sampled from the fine-tuning dataset.

**Co-FT** : We fine-tune all parameters using the behavioral cloning loss, and each update batch is sampled from both the pretraining dataset and the finetuning dataset with a fixed weight. See Section A.8 for specific weight values we use for different tasks.

**LoRA** : LoRA (low rank adaptation) freezes all the weights of the base pretrained policy, and finetunes an adapter head with a low rank bottleneck. Typically the adapter head has much fewer parameters than the base pretrained model. The resulting policy is achieved by adding the weights of the frozen pretrained policy and the low rank adapter head.

**Freeze-FT** : Similar to Task-FT, but we freeze the parameters in the language model backbone and finetune only parameters from the action expert and vision encoder.

**Scratch** : Training a policy from sratch. To make it comparable to the other VLA baseline policies, we use the same $\pi_0$ architecture but initialize from the Paligemma VLM weights, without pretraining on any robot data.

### A.10 Why does RETAIN work so well?

While we have shown empirically in this work that RETAIN works well across real and simulated tasks, we don't understand the full scope of the reasons why model parameter merging works so well empirically. However, previous work in computer vision and large-language models have also shown empirical benefits of merging parameters of the pretrained and fine-tuned model (see Section 2). Similar to our work, these previous works are also largely empirical and corroborate our findings in a real-world robotics setting. Specifically, Neyshabur et al. (2020) found that fine-tuning from the same pretrained model results in regions where solutions are connected by a linear path along which error remains low, a phenomenon known as "linear mode connectivity" [2]. [3] and [4] explained that SGD typically converges to a solution that is on the boundary of this low-error path, while weight merging is able to find a point centered in this region, which often has slightly worse train loss but substantially better test error. We attribute the performance gains we see also to this, though call for more rigorous future work to explain this more rigorously.

### A.11 Qualitative Analysis of Success and Failure Mode of RETAIN

Typically, we observe that RETAIN improves the robustness of the merged policy on OOD evaluations. Compared to task-FT policies, which are brittle and will fail in out-of-distribution scenarios catastrophically and is unable to retry, the RETAIN policies typically exhibits more robust behavior, and can recover from failure using its generalist knowledge. Typically, we observe that the task-FT policy either either does the full task successfully, or cannot do the task at all. In comparison, the RETAIN policies usually at least partially complete the task. However, we do observe that the RETAIN policies sometimes fail due to (1) imprecise execution of the task and stuck in constant retry mode and (2) produces an action that does not solve the task (though still semantically meaningful), and is unable to successfully continue afterwards. We provide two qualitative examples of this in Fig. 24 and Fig. 25.

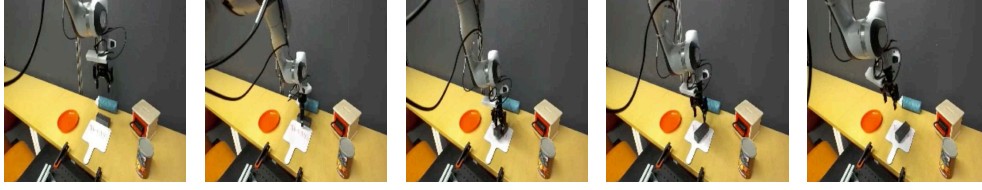

Figure 24: Failure Example of RETAIN in DROID `whiteboard` task: The arm picks up the eraser, but drops it on the whiteboard instead of wiping left and right with it.

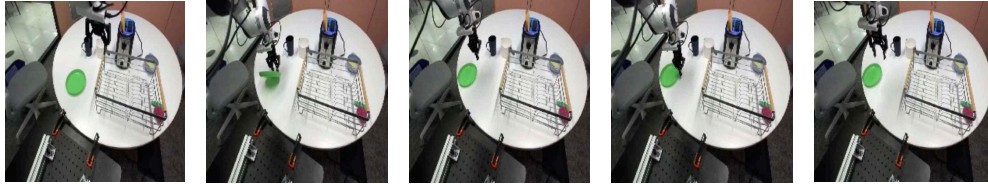

Figure 25: Failure Example of RETAIN in DROID `plates` task: the arm is not able to precisely pick up the green plate, and so constantly retries this until the policy times out.

