# OpenReview forum: "Robust Fine-tuning of Vision-Language-Action Robot Policies via Parameter Merging"
_ICLR.cc/2026/Conference — ICLR 2026 Poster_

### Official Review · Reviewer_Rx6M · 2025-10-16

**Soundness:** 3
**Presentation:** 4
**Contribution:** 3
**Rating:** 8
**Confidence:** 4

**Summary:**

This paper addresses a critical challenge in robot learning: finetuning large, generalist Vision-Language-Action (VLA) policies on new tasks with limited demonstration data often leads to catastrophic forgetting of prior skills and poor generalization on the new task itself. The authors propose a surprisingly simple yet effective method called RETAIN (Robust finE-tuning with parameter mergINg). The core idea is to linearly interpolate the weights of the pretrained generalist policy and the policy after it has been finetuned on a small dataset for the new task. The authors evaluate RETAIN across both simulated (LIBERO) and real-world (DROID) environments, using a pretrained π₀-style policy. Their evaluation framework rigorously distinguishes between three settings: 1) in-distribution (ID) performance on the new task, 2) OOD performance on variations of the new task, and 3) performance on the original "generalist" tasks. The results demonstrate that RETAIN significantly improves OOD generalization on the new task compared to standard finetuning methods (Task-FT, Co-FT, LoRA) while successfully retaining the policy's original generalist capabilities. The paper further analyzes modality-specific merging, finding that merging the language model parameters is often most critical , and shows initial promise for using RETAIN in a sequential, continual learning setting.

**Strengths:**

- The proposed method is exceptionally simple to implement, requiring only a weighted average of two model checkpoints. This lack of complexity is a major advantage for practical application, as it incurs no additional training or inference cost and can be easily integrated into existing workflows.
- The paper tackles a highly relevant and pressing issue. As the field moves towards large foundation models, the ability to efficiently and robustly adapt them to new, specific tasks with minimal data is a key bottleneck for real-world deployment. The paper's motivation is strong and clearly articulated with empirical evidence of overfitting in standard methods (Figure 2).
- The experimental design is a significant strength. The use of both simulation (LIBERO) and real-world robots (DROID) provides compelling evidence. The explicit differentiation between ID, OOD, and Generalist performance allows for a nuanced analysis that directly supports the paper's core claims about improving robustness and retaining prior knowledge. This evaluation structure is a practical application of the principles advocated by more formal taxonomies like STAR-Gen.
- The finding that the LLM backbone parameters are the most influential for merging  provides a deeper insight into the functioning of VLAs. It suggests that the generalizable, abstract knowledge is heavily encoded in the language-centric weights, which aligns with the broader understanding that VLM pretraining is a primary source of semantic reasoning.

**Weaknesses:**

- While the paper demonstrates that parameter merging works, it offers limited insight into why it works so effectively in the non-convex loss landscape of deep neural networks. The approach is presented as an empirical observation. A more rigorous discussion, potentially drawing from literature on model soups, loss landscapes, and mode connectivity in NLP/CV, would significantly strengthen the paper. For instance, does merging find a wider, flatter minimum between the pretrained and finetuned solutions, which is known to correspond to better generalization? This lack of theoretical grounding leaves the method feeling more like a "trick" than a principled technique.
- The merging coefficient $\alpha$ is the method's key hyperparameter, and its selection is not adequately addressed. The authors state they swept over values to find the best-performing ones for their experiments. This is not practical for a real-world user who may not have the resources to perform extensive OOD evaluation to tune $\alpha$ for every new task. How should a user select $\alpha$ without access to a diverse OOD validation set? The paper does not propose a heuristic or a low-cost method for estimating a good value. The optimal $\alpha$ values differ significantly between the LIBERO (often 0.8-0.9) and DROID (0.5-0.75) experiments. The paper attributes lower OOD improvement in LIBERO to a less capable base model, but does not explain why this would lead to a different optimal merging ratio. This suggests that the optimal $\alpha$ may depend on the pretrained model's quality, the finetuning dataset size, and the task similarity, making it difficult to set a priori.
- The paper claims that RETAIN "enables continual acquisition of new skills in a lifelong learning setting". However, the supporting experiment involves only two sequential tasks. This is insufficient evidence for robust continual learning.

**Questions:**

- The paper reports success rates but lacks a qualitative analysis of failure cases. When the RETAIN policy fails, how does its failure mode differ from the base policy or the fully finetuned policy? Does it produce more conservative, "safer" failures by reverting to the generalist model's behavior, or does it produce novel, unpredictable failures stemming from the weight interpolation? This analysis would provide valuable insight into the method's reliability.
- How does the method perform after 5, 10, or N tasks? Does performance on earlier tasks degrade as more skills are merged? The proposed iterative merging process ($\tilde{\theta}_{n}=(1-\alpha)\cdot\tilde{\theta}_{n-1}+\alpha\cdot\theta_{ft,n}$) might dilute the knowledge of the original pretrained model and early tasks over time.
- A key finding from the STAR-Gen benchmark was that VLAs struggle significantly with semantic generalization (e.g., novel instructions for the same behavior). Does RETAIN improve robustness to linguistic variations, or does it primarily bolster visual robustness? How does the method handle changes to unobservable physical properties (e.g., mass, friction), another axis in STAR-Gen?
- Does a larger or more diverse finetuning dataset generally permit a higher $\alpha$ (leaning more towards the specialist model), whereas a smaller, narrower dataset benefits from a lower $\alpha$ to better preserve the generalist priors?
- Your choice of linear interpolation is compelling due to its simplicity. I am curious whether you experimented with or considered other parameter merging techniques from the broader deep learning literature. For example, methods like task arithmetic (where task vectors are computed and added/subtracted) have shown success in NLP. Was linear interpolation the first and most obvious choice, or did you find it superior to other, perhaps more complex, merging schemes for VLA policies?

---

> ### Author Response · Authors · 2025-11-21
> **Overview & more insights into why parameter merging works**
>
> Thank you for your time and your review! We are glad to hear that you think our paper addresses a critical problem with a simple and efficient method, and found our experiments well designed and compelling, and the results insightful. We provide answers to your questions below:
>
> > While the paper demonstrates that parameter merging works, it offers limited insight into why it works so effectively in the non-convex loss landscape of deep neural networks.
>
> Great point! As you pointed out, while our paper is largely empirical and does not strive to uncover a full scope understanding of why RETAIN is so effective, we want to provide **additional ablation and analysis experiments** for a better understanding of the method. In Appendix A.3, we added new experiments and found that the **finetuning path is highly non-linear in parameter space and model merging achieves a different solution compared to finetuning**. In addition, we added discussions in Appendix A.8, drawing from model merging works in NLP/CV, to explain why model merging works so well. We will explain each one of these in more detail below.
>
> The new Appendix A.3 seeks to understand how RETAIN works by asking the following research questions: What does the finetuning path look like in parameter space during normal supervised finetuning? Where does the parameter-merged model lie in the parameter space compared to the finetuned models? In Appendix A.3, we plot three different visualizations of how the finetuned model evolves in parameter space, and all visualizations show that the finetuning path is highly non-linear. Here, we will highlight one of the three visualization methods; please refer to A.3 for all details. We visualize the path of the difference vectors $\theta_{i+1} - \theta_{i}$, where $\theta_i$ the parameter of the fine-tuned checkpoint during the process of fine-tuning at every 100 steps. We then project this difference vector to 2D using principal component analysis (PCA), and visualize the trajectory in 2D. If the finetuning parameter path was linear, the PCA projection would show a straight line since all difference vectors should be aligned in direction. However, Fig 15 shows that the finetuned path (blue lines) is highly non-linear. We also plot the parameter of the merged model in this same PCA-projected space, and find that the merged model ends up at a different spot than the fine-tuned model. This shows that the model merging solution ends up to be quite different from the finetuned model.
>
> In Appendix A.8, we relate to model-merging work in CV/NLP to explain why model merging works so effectively. Previous work in computer vision and large-language models have also shown empirical benefits of merging parameters of the pre-trained and fine-tuned model (see related work section). Similar to our work, these previous works are also largely empirical and corroborate our findings in a real-world robotics setting. Specifically, [1] found that fine-tuning from the same pre-trained model results in regions where solutions are connected by a linear path along which error remains low, a phenomenon known as “linear mode connectivity” [2]. [3] and [4] explained that SGD typically converges to a solution that is on the boundary of this low-error path, while weight merging is able to find a point centered in this region, which often has slightly worse train loss but substantially better test error. We attribute the performance gains we see also to this.
>
> [1] Neyshabur, Behnam, Hanie Sedghi, and Chiyuan Zhang. "What is being transferred in transfer learning?." Advances in neural information processing systems 33 (2020): 512-523.
>
> [2] Frankle, Jonathan, et al. "Linear mode connectivity and the lottery ticket hypothesis." International Conference on Machine Learning. PMLR, 2020.
>
> [3] Izmailov, Pavel, et al. "Averaging weights leads to wider optima and better generalization." arXiv preprint arXiv:1803.05407 (2018).
>
> [4] Wortsman, Mitchell, et al. "Robust fine-tuning of zero-shot models." Proceedings of the IEEE/CVF conference on computer vision and pattern recognition. 2022.

---

> ### Author Response · Authors · 2025-11-21
> **Hyperparameter selection, failure cases, and others**
>
> > The merging coefficient  is the method's key hyperparameter, and its selection is not adequately addressed
>
> We updated Appendix A.6 to clearly detail how we choose the merging coefficient hyperparameter $\alpha$. We also explain it below here:
>
> For each task that we consider (e.g. mugs-on-plate in LIBERO, plates in DROID), we test policy performance on several scenes, which are different variations of the same task with different objects, distractors, and backgrounds etc. Our hyperparameter section process is: we select $\alpha$ based on its OOD evaluation performance on one held out validation scene, and then we use the same $\alpha$ to report performance on several test scenes. We do not tune $\alpha$ for the different test scenes. So in practice, we only need to tune this parameter in one scene, and it can do well in different scenes. While we do pick $\alpha$ differently per task, which is the common practice for hyperparameter tuning in ML literature, we actually find that $\alpha$ values can be robust to different tasks. For example, in our DROID experiments, we only tune from values 0.25, 0.5, and 0.75, and find that $\alpha=0.5$ tends to perform well across almost all tasks. The LIBERO experiments use a different range of $\alpha$ values (often 0.8 or 0.9) because it is a different domain with different pre-trained models. But the optimal $\alpha$ values is often similar within the LIBERO setting.
>
> > The paper reports success rates but lacks a qualitative analysis of failure cases.
>
> We added Appendix A.10 to qualitatively analyze the success and failure modes of RETAIN. In essence, we observe that RETAIN typically performs better than task-FT in out-of-distribution scenarios because it produces more reasonable actions toward solving an OOD task, and can produce retry behavior when failing for the first time. In comparison, the task-FT policy either can near-fully complete the task, or cannot do the task at all. We observe that RETAIN policies can fail due to (1) imprecise execution of the task and stuck in constant retry mode and (2) producing an action that does not solve the task (though still semantically meaningful), and is unable to successfully continue afterwards. We provide two example timestrips in Fig 21 and 22.
>
> >  Does RETAIN improve robustness to linguistic variations, or does it primarily bolster visual robustness? How does the method handle changes to unobservable physical properties (e.g., mass, friction), another axis in STAR-Gen?
>
> We did not explore semantic and linguistic generalization, nor generalization to unobservable physical properties. In this work, we mainly focused on generalization to different visual variations (lighting conditions, camera angle, background), scene variations (distractor objects, table height), and action variations (different shape and texture of objects, different grasping angles). We agree that the different types of generalization you pointed out is an interesting area for future work.
>
> > Does a larger or more diverse finetuning dataset generally permit a higher $\alpha$ (leaning more towards the specialist model), whereas a smaller, narrower dataset benefits from a lower $\alpha$ to better preserve the generalist priors?
>
> We have not observed this. In our DROID experiments, the plates task had 100 demonstrations while the whiteboard task only had 50 demonstrations. While the two datasets are different sizes, we generally find $\alpha=0.5$ to work well across both tasks.
>
> >  I am curious whether you experimented with or considered other parameter merging techniques from the broader deep learning literature
>
> We did experiment with a few other alternative merging methods (task vector arithmetic, spherical linear interpolation [1], drop-and-rescale [2]) but did not find that they have obvious advantages over simply linearly interpolating the weights. Since those experiments were preliminary (e.g. it may require more careful tuning), we did not include them in the paper. The linear interpolation is also desirable because of its simplicity and only has one hyperparameter. We do think that a smarter merging scheme is a good area for future research.
>
> [1] Barrera, Tony, Anders Hast, and Ewert Bengtsson. "Incremental spherical linear interpolation." The Annual SIGRAD Conference. Special Theme-Environmental Visualization. Vol. 13. Linköping, Sweden: Linköping University Electronic Press, 2004.
>
> [2] Yu, Le, et al. "Language models are super mario: Absorbing abilities from homologous models as a free lunch." Forty-first International Conference on Machine Learning. 2024.

---

> > ### Comment · Reviewer_Rx6M · 2025-11-21
> >
> > Thanks for the authors' detailed response. I will keep my score.

---

### Official Review · Reviewer_mT9g · 2025-10-26

**Soundness:** 3
**Presentation:** 4
**Contribution:** 2
**Rating:** 6
**Confidence:** 3

**Summary:**

This work focuses on VLA fine-tuning, and proposes model merging to induce robustness and mitigate forgetting. Standard fine-tuning on reasonably-sized single-task datasets (i.e. ~100 demonstrations) are known to improve performance on the specific tasks, but degrade it elsewhere (as confirmed by the authors in Figure 2); standard solutions (such as co-training or LORA) tend to only marginally alleviate this issue, or require access to the pre-training dataset. The authors propose a simple linear interpolation strategy between pre-trained an fine-tuned weights, which may optionally specialize to the three sub-networks (vision encoder, language backbone and action head). This technique can be applied in a single fine-tuning round, or over multiple rounds in a continual learning setting. Authors claim that model merging preserves the generalist performance of the VLA, while also boosting OOD performance on the fine-tuning tasks. Their claims are supported by experiments spanning real (DROID) and simulated (LIBERO) environments. The authors include a straightforward ablation of the interpolation parameter $\alpha$.

**Strengths:**

- Experimental evaluation is extensive, involving both simulated and real-world experiments. Moreover the experiments are described in great detail.
- The problem tackled by this work is timely has clear practical interest.
- This paper is written in a clear and understandable way.
- The evaluation of different merging coefficient for different components is interesting on its own.

**Weaknesses:**

- The main weakness of this work is that it is not entirely clear whether the overfitting issue arises due to sub-optimal hyperparameter choices. Figure 2 shows that performance of baselines such as Co-FT fluctuates and decays for a large number of gradient steps: at 500 gradient steps both ID and OOD performance on the new task is high, and generalist performance has not decayed. The supplementary material details that the number of gradient steps was chosen for each algorithm/environment combination, but the tuning process is not clear.
- The performance gains from model merging are often minor, and the method does not consistently beat baselines such as Co-FT (e.g. in continual learning settings). More importantly, these performance gains might be due to precise tuning of $\alpha$, which is set on a per-task basis. It is unclear how well the method would perform out of the box, without extensive tuning of $\alpha$, which might be prohibitive in real-world settings.

A few minor issues will be detailed in the following section.

**Questions:**

### Minor issues
- Model merging is not an entirely novel idea, as the authors acknowledge in the related works. I would suggest adding a couple of references in the introduction as well, which does not point to previous work in model merging. This is not a major weakness, as the authors acknowledge existing works, and focus on empirical evaluations in a multi-modal setting, which is to the best of my knowledge under investigated.
- Line 143: dataset tuples should be $(s_t^{(i)}, a_t^{(i)}, T^{i})$ to be consistent with the notation in Equation 1.
- Line 196 is later contradicted: the model does not improve monotonically the longer it is trained, even ID.
- "mugs" is not formatted correctly in the caption of Table 14.

### Questions
- Did the authors consider regularization to the pre-trained model (i.e. distillation) as an option to prevent model drift? This is for instance done to prevent response collapse when fine-tuning LLMs.
- What do error bars represent in Figures 6 and 7?

### Conclusion
Overall, the paper makes an interesting contribution to an open problem, and I currently recommend acceptance. My main concerns revolve around the experimental settings, and the modest gain in performance with respect to Co-FT. These concerns are outweighed by the fact that (a) experiments are well described, and thus different training regimes can be explored in future work and (b) the strongest baselines require access to pre-training data, while model merging does not. Nonetheless, clarifying whether the training regimes considered are already designed to avoid as much overfitting as possible would improve this work's significance.

---

> ### Author Response · Authors · 2025-11-21
> **Overview**
>
> Thank you for your time and your review! We are glad to hear you appreciated the value of our method and liked our extensive experiments and clear writing. In the following, we (1) add new ablation experiments and detail our tuning process to show that overfitting is not fixed by hyperparameter tuning and (2) add new real world DROID experiments with four new OOD scenes, and explain the performance gains over baselines.
>
> Please let us know if you have any additional concerns and we are happy to discuss!

---

> ### Author Response · Authors · 2025-11-21
> **Hyperparameter choices**
>
> > “The supplementary material details that the number of gradient steps was chosen for each algorithm/environment combination, but the tuning process is not clear.”
>
> We choose the number of gradient steps per-task by picking the **earliest checkpoint that achieves the best success rate in ID evaluation** during fine-tuning. We use this principle to pick checkpoints because it (1) can fully fit the finetuning demonstrations and do well in ID evaluations and (2) is not overfitted yet and be the most competitive baseline for OOD and generalist evaluation. We have updated this in Appendix A.6.
>
> > “The main weakness of this work is that it is not entirely clear whether the overfitting issue arises due to sub-optimal hyperparameter choices”.
>
> Great question! To investigate this, we provide new experiment results in Appendix A.2 (highlighted in red) to ablate different choices of learning rate and the number of gradient steps. The question we aim to answer with these new experiments is whether finetuning with a lower learning rate or smaller number of gradient steps fixes the overfitted issue.
>
> We plotted three types of evaluation performances (ID, OOD, Generalist) in Fig 12 and 13, where each subplot shows a different learning rate (one bigger than what we used in the paper, and two smaller), and we evaluate each model at every 100 gradient steps to also ablate the number of update steps. With a larger learning rate, it’s clear that the overfitting issue is more severe, and the performance on all three kinds of evaluations (ID, OOD, Generalist) go down to near zero. With a smaller learning rate, the model still performs well in ID evaluations, and suffers less from forgetting generalist capabilities (measured from Generalist evaluations). This is to be expected because the finetuned model is closer to the pre-trained model with a smaller learning rate. However, note that with a smaller learning rate, the OOD evaluation performance is also worse. We plot the OOD performance achieved using the original learning rate in the paper as the dotted orange line in Fig 12 & 13, and the gap between the dotted and solid orange line shows the performance gap in OOD evaluation when we lower the learning rate, possibly due to underfitting. This shows that lowering the learning rate or number of update steps does not achieve the same effect as model merging.
>
> **This new experiment shows that lowering the learning rate and number of gradient steps does not solve the problem of overfitting during finetuning, and can cause underfitting.**
>
> > These performance gains might be due to precise tuning of $\alpha$, which is set on a per-task basis. It is unclear how well the method would perform out of the box, without extensive tuning of $\alpha$.
>
> **We do not extensively tune $\alpha$ in our experiments**. In fact, in the DROID experiments where RETAIN achieves the strongest result, we only tested three different values of $\alpha \in$ (0.25, 0.5, 0.75) in one scene, and find that $\alpha=0.5$ almost always perform well on all scenes across all tasks. We use the same $\alpha$ value in our new DROID experiments with four new OOD scenes **without any tuning**, and the updated Fig 6 shows that this **same hyperparameter value still works well** and significantly outperform baselines. This shows that RETAIN is robust to this hyperprameter choice and can work well without extensive tuning. We included a new section in Appendix A.6 for more discussion on our hyperparameter tuning process.

---

> ### Author Response · Authors · 2025-11-21
> **Performance gains and others**
>
> > The performance gains from model merging are often minor, and the method does not consistently beat baselines such as Co-FT (e.g. in continual learning settings)
>
> We want to highlight that the performance gain over baselines is actually significant, especially in our real world DROID experiments. To further test how well RETAIN perform in real world settings, we added **four new OOD scenes** for our two DROID tasks with **240 evaluations trials in total**. See Figure 5 for the updated visualizations of these scenes.
>
> The trend of the evaluation result on these four new scenes aligns with our findings in the original submission, and we have updated Fig 6 to reflect the new results and shows the per-task breakdown. We find that RETAIN methods outperform baselines at OOD evaluations. In addition, RETAIN methods also perform the best in Generalist evaluation out of any finetuned model, matching the performance of the pre-trained generalist model.
>
> We have also updated our continual learning results in Fig 10 after evaluating the four new OOD scenes.  **RETAIN outperforms co-FT in the sequential setting under all tasks and evaluation types.**
>
> > RE: Minor issues (on writing)
>
> Thanks for pointing these out, we have fixed these issues (in red) in the revised submission. We have also added references for previous model merging work in the Introduction section, as you suggested.
>
> > Did the authors consider regularization to the pre-trained model (i.e. distillation) as an option to prevent model drift?
> This is for instance done to prevent response collapse when fine-tuning LLMs.
>
> Thanks for the suggestion! We have not tried using distillation as a regularization method to the pre-trained model; to the best of our knowledge, this is under-explored in fine-tuning robotic foundation models and is a good area of future work. Our method, RETAIN, can also be seen as a form of regularization to the pre-trained model, as it actually inherits interpolated weights from the pre-trained model to prevent model drift. However, as we have shown, RETAIN is more than a regularization method and actually helps transfer pre-trained knowledge and apply it to better solve OOD tasks.
>
> > What do error bars represent in Figures 6 and 7?
>
> The error bars represent standard error.

---

> > ### Comment · Reviewer_mT9g · 2025-11-21
> >
> > Thank you for addressing my comments thoroughly. I have updated my score accordingly.

---

### Official Review · Reviewer_Jiug · 2025-10-27

**Soundness:** 3
**Presentation:** 4
**Contribution:** 4
**Rating:** 8
**Confidence:** 4

**Summary:**

This paper addresses the overfitting of generalist robot policies during small-scale dataset finetuning, a process that compromises pre-trained knowledge and inhibits generalization to OOD tasks. The proposed method, RETAIN, mitigates this by merging the parameter spaces of the pre-trained and finetuned policies via linear interpolation. Experimental results indicate that RETAIN achieves robust generalization to novel tasks while preserving broad, general capabilities. The paper also provides a comparative analysis against prior strategies, offering specific insights into weight merging with different parameter groups.

**Strengths:**

- The paper addresses a critical and practical problem in robotics: how to adapt generalist policies to new tasks with limited data without overfitting or forgetting.
- The proposed method, RETAIN, is simple and efficient, relying on parameter interpolation with no added inference cost.
- The authors provide a valuable comparative analysis against several prior finetuning strategies, offering significant insights to the research community.
- The evaluation protocol is clear and effectively highlights the core problem.

**Weaknesses:**

1. The selection of the $\alpha$ merging coefficient is unclear and appears to be highly task-dependent (from Appendix A.5). The paper does not offer a methodology or heuristic for tuning $\alpha$ based on specific task characteristics.
2. The insightful analysis on modality-specific merging (Sec 6.5) seems limited to simulation experiments. It is unclear if the conclusion that only LLM parameters need merging holds for the more complex, real-world DROID tasks.
3. The continual learning claims (Sec 6.6) are partially undermined by an unexplained anomaly in Figure 10, where the co-FT baseline achieves surprisingly high OOD performance on the second task, which the authors acknowledge but do not investigate.

**Questions:**

- For weakness 1: What is the recommended procedure for selecting $\alpha$ in a practical? Can $\alpha$ be estimated from the data?
- For weakness 2: Was the modality-specific merging analysis (i.e., merging only LLM parameters) validated on the real-world DROID tasks? Does the same finding hold?
- For weakness 3: Do the authors have a hypothesis for the anomalous co-FT result in the continual learning experiment (Fig 10)? Could this be due to a specific relationship between the $\textit{plates}$ and $\textit{whiteboard}$ tasks?

---

> ### Author Response · Authors · 2025-11-21
> **Author Response**
>
> Thank you for your time and your review! We are glad to hear that you think our paper addresses a critical problem with a simple and efficient method, and found our experiment protocol clear and effective, and the results insightful. We provide answers to your three questions below:
>
> > The selection of the  merging coefficient is unclear and appears to be highly task-dependent (from Appendix A.5). The paper does not offer a methodology or heuristic for tuning  based on specific task characteristics.
>
> We updated Appendix A.6 to include more details about how we selected the hyperparameter $\alpha$ in practice. While this parameter is task dependent, we found in our real world DROID experiments that it is actually quite robust because we only had to do slight tuning among the values 0.25, 0.5, and 0.75. On almost all tasks, $\alpha=0.5$ performs well.
>
> In our experience, the best value of $\alpha$ depends on how out-of-distribution the test scene is to the pre-training data and to the finetuning demonstration dataset. When the test scene is close to the finetuning dataset, higher values of $\alpha$ is typically better, while when the test scene is very far from the finetuning data lower values of $\alpha$ is better. However, it is in practice very challenging to measure how out-of-distribution a test scene is to the finetuning dataset and to the pretraining dataset. If such a heuristic exists, $\alpha$ can be chosen based on that estimate. We think this is an important area for future work.
>
> >  Was the modality-specific merging analysis (i.e., merging only LLM parameters) validated on the real-world DROID tasks? Does the same finding hold?
>
> We did not conduct modality-specific merging on real-world DROID tasks because of limitations of the pretrained model. This is because we chose the Pi0-FAST-DROID as the pre-trained model for our DROID experiments, since it was the best open-source DROID policy at the time of our experiments as judged by the RoboArena policy ranking [1]. This base policy does not include an action expert, and so effectively the actions are produced by the language modality in the LLM backbone. Therefore, we are not able to perform modality-specific merging experiments shown in Fig 9. We did try the Pi0-DROID base model, which does include an action expert and therefore an action modality, but found it performs poorly on our DROID tasks, corroborating findings by [1].
>
> [1] Atreya, Pranav, et al. "RoboArena: Distributed Real-World Evaluation of Generalist Robot Policies." arXiv preprint arXiv:2506.18123 (2025).
>
> > The continual learning claims (Sec 6.6) are partially undermined by an unexplained anomaly in Figure 10, where the co-FT baseline achieves surprisingly high OOD performance on the second task, which the authors acknowledge but do not investigate.
>
> We added four new OOD scenes (two for the plates task, and two for the whiteboard task, see Fig 5 for new visualizations) for our DROID experiments and tested our continual learning methods on these new OOD scenes. We updated Fig 10 to show the average success rates on all the OOD scenes. As expected, now the co-FT baseline achieves low performance on average on the second task. This new results shows that RETAIN performs better than co-FT in the sequential setting under all tasks and evaluation types. We think the original anomaly is caused by some intricate learning dynamics during lifelong learning.

---

> > ### Comment · Reviewer_Jiug · 2025-11-22
> >
> > Thanks to the authors for answering my question. I will keep my score.

---

### Official Review · Reviewer_R6dX · 2025-10-30

**Soundness:** 2
**Presentation:** 4
**Contribution:** 3
**Rating:** 4
**Confidence:** 4

**Summary:**

The paper proposes a very simple strategy for finetuning of policies by merging model parameters of pretrained models and finetuned models. The work presents empirical results on Droid robot setup and Libero simulations and find a surprising effectivity of this method in achieving task performance in the fine tuned task while retaining performance on the original (generalists) tasks as well as generalization to variations of the fine-tuning task.

**Strengths:**

- simple method
- extensive experiments

**Weaknesses:**

- no deeper understanding
- same unclear results and overly strong claims
- questionable hyperparameter selection

Because the method is so simple and not really clear, from a fundamental perspective, why it should have the presented favorable properties, extra care has to be taken in conducting the experiments and interpreting the results. See my questions below.

**Questions:**

I know that your paper is not trying to give a deeper understanding why the simple model merging does the right thing. Nevertheless, I would like to know if you have analyzed the following:
- we do a linear interpolation in weight space.
- if the fine-tuning training path in weight space would be more or less linear, then simply training with a lower learning rate or less steps should have the same effect.
Here the actual questions:
Q1a: Have you ablated for learning rate / number of updates steps.
Q1b: I think a non-linearity check for the gradient path would be interesting: compare the vector of $\theta_0$ to $\theta_{n/2}$ with $\theta_{n/2}$ to $\theta_{n}$ in terms of collinearity.

Q2: The results in the paper appear cherry picked in comparison to all results, e.g. presented in Fig. 12.
For those tasks, Co-FT and RETAIN-Co-FT are very similar here (sometimes one or the other wins).
I suggest creating a more balanced representation in the main paper and tone down your claims.

Q3: Sec. 6.5: How was the $\alpha$ parameter selected?  I understood that $\alpha$ (I guess the same for all modalities) is selected based on the best OOD test performance. This is against the Machine Learning practice! Never select hyperparameters on the test set! Maybe I am mistaken here, please clarify.

Q4: Which $\alpha$ parameter did you use for the DROID experiments? How was it selected?
Also, did you use different $\alpha$s for the modalities in the experiments, and how did you select this weighting?

Q5: How does alpha depend on training parameters (how many updates) etc?

Q6: Sec 6.6: What would happen if you finetune on both / all sequential tasks at once. As you do Co-FT, you anyway have all the data available, so I think that would be a good comparison / upper baseline.

Q7: for the sequential learning, what would you expect if one finetunes the base model on both tasks and then merges both together, as a weighted sum. This would allow parallel improvement and sounds like an interesting use case.

Comments:
- You have one or three additional hyperparameters (see Q4), and depending on their selection, I am not so surprised to see that you can find a setting that works well in your experiments, the question is: will these hp settings also work for a new environment?

Related work: also a paper that looks into finetuning from few demonstrations:
Active Fine-Tuning of Multi-Task Policies, Bagatella et al
This paper does not do model merging but learned output merging and might be interesting to related to.

---

> ### Author Response · Authors · 2025-11-21
> **Overview**
>
> Thank you for your time and your review! To address your comments, we provide below (1) some new analysis experiments to achieve a mechanistic understanding of RETAIN, as well as explanation from previous parameter-merging work on why it helps generalization. (2) We clarify that the main results shown are averaged across all tasks and are not cherrypicked. (3) Finally, to your concern on selecting parameters on test-set performance, we clarify our hyperparameter selection process, and present new results on four new OOD DROID scenes.
>
> Are there additional questions we could address to improve the score?

---

> ### Author Response · Authors · 2025-11-21
> **Analysis of why RETAIN / model parameter merging works [part 1]**
>
> > Q1: not really clear, from a fundamental perspective, why it should have the presented favorable properties.
>
> Great question! As you have pointed out in your review, the goal of our paper is not to give a full scope understanding of model parameter merging. Regardless, we provide a few sets of additional ablation and analysis experiments below to provide better understanding of the RETAIN method. We find that **(1) using a smaller learning rate or a smaller number of gradient steps does not prevent overfitting and does not achieve the same effect as parameter merging** and **(2) the fine-tuning path in weight space is highly non-linear and therefore parameter merging results in different final models than the finetuned ones**. Finally, we also include some explanations from previous model merging work on why it helps. See our detailed response below.
>
> > Q1a: Have you ablated for learning rate / number of updates steps
>
> We added a new ablation of this in Appendix A.2, where we plotted three types of evaluation performances (ID, OOD, Generalist) in Fig 12 and 13. Each subplot shows a **different learning rate (one bigger than what we used in the paper, and two smaller)**, and we **evaluate each model at every 100 gradient steps to also ablate the number of update steps**. With a larger learning rate, it’s clear that the overfitting issue is more severe, and the performance on all three kinds of evaluations (ID, OOD, Generalist) go down to near zero. With a smaller learning rate, the model still performs well in ID evaluations, and suffers less from forgetting generalist capabilities (measured from Generalist evaluations). This is to be expected because the finetuned model is closer to the pre-trained model with a smaller learning rate. However, note that with a smaller learning rate, the OOD evaluation performance is also worse. We plot the OOD performance achieved using the original learning rate in the paper as the dotted orange line in Fig 12 & 13, and the gap between the dotted and solid orange line shows the performance gap in OOD evaluation when we lower the learning rate, possibly due to underfitting. This shows that **lowering the learning rate or number of update steps does not achieve the same effect as model merging**.
>
> > Q1b: I think a non-linearity check for the gradient path would be interesting: compare the vector of $\theta_0$ to $\theta_{n/2}$ with $\theta_{n/2}$ to $\theta_n$ in terms of collinearity.
>
> Thanks for the suggestion! We added Appendix A.3 to specifically discuss the gradient/parameter path during fine-tuning and during model parameter merging.
>
> We plotted your suggested experiment in Figure 14, where we check the collinearity of vectors $\theta_{i+1} - \theta_{i}$ and $\theta_{i} - \theta_{i-1}$, where $\theta_i$ the parameter of the fine-tuned checkpoint during the process of fine-tuning. We measure the collinearity as the cosine similarity between the two vectors, and plot every 100 steps during fine-tuning. Fig 14 shows that the **changes in parameter space is highly non-linear since the cosine similarity values are not close to 1**. This is to be expected since the parameter trajectory of deep neural networks is often highly non-linear.
>
> In addition, we also **plot in Fig 15 a PCA projection of the parameter difference ($\theta_{i+1} - \theta_{i}$) to a 2D space**. This is another more intuitive visualization of the parameter/gradient path during fine-tuning. If the parameter path was linear, the PCA projection should show a straight line since all difference vectors should be aligned in direction. Therefore, Fig 16 also shows that the finetuning path is highly non-linear. Furthermore, we also plot the parameter of the merged model in the same PCA-projected space, and find that the merged model ends up at a different spot than the finetuned models. We also generalize this visualization of the two principal directions to all directions, by visualizing the singular values of the difference vector matrix in Fig 16. If the difference vectors lie in a linear path, then all difference vectors would point in the same direction and there should only be 1 non-zero singular value. However, it's clear Fig 16 that most singular values are non-zero, showing that the path is non-linear in many dimensions. See more details in Appendix A.3.
>
> All of these new analysis experiments show that the **fine-tuning path is highly non-linear, and therefore model parameter merging actually results in a different solution than the finetuned models**.

---

> ### Author Response · Authors · 2025-11-21
> **Analysis of why RETAIN / model parameter merging works [part 2]**
>
> > some explanations from previous model merging work on why it helps
>
> Previous work in computer vision and large-language models have also shown empirical benefits of merging parameters of the pre-trained and fine-tuned model (see related work section). Similar to our work, these previous works are also largely empirical and corroborate our findings in a real-world robotics setting. Specifically, [1] found that fine-tuning from the same pre-trained model results in regions where solutions are connected by a linear path along which error remains low, a phenomenon known as “linear mode connectivity” [2]. [3] and [4] explained that SGD typically converges to a solution that is on the boundary of this low-error path, while weight merging is able to find a point centered in this region, which often has slightly worse train loss but substantially better test error. We attribute the performance gains we see also to this, though call for more rigorous future work to explain this more rigorously. We have also added this discussion to Appendix A.8.
>
> [1] Neyshabur, Behnam, Hanie Sedghi, and Chiyuan Zhang. "What is being transferred in transfer learning?." Advances in neural information processing systems 33 (2020): 512-523.
>
> [2] Frankle, Jonathan, et al. "Linear mode connectivity and the lottery ticket hypothesis." International Conference on Machine Learning. PMLR, 2020.
>
> [3] Izmailov, Pavel, et al. "Averaging weights leads to wider optima and better generalization." arXiv preprint arXiv:1803.05407 (2018).
>
> [4] Wortsman, Mitchell, et al. "Robust fine-tuning of zero-shot models." Proceedings of the IEEE/CVF conference on computer vision and pattern recognition. 2022.

---

> ### Author Response · Authors · 2025-11-21
> **Results clarification, hyperparameter selection process, and others**
>
> > Q2: The results in the paper appear cherry picked in comparison to all results, e.g. presented in Fig. 12. For those tasks, Co-FT and RETAIN-Co-FT are very similar here (sometimes one or the other wins).
>
> We want to clarify that we **in no way cherry picked the results; all results are presented in the main paper**. Specifically, Fig 7 shows the averaged result in LIBERO, which is averaged over ALL tasks shown in Fig 11 (originally number Fig 12 in original submission), with no cherrypicking. While Co-FT is a bit better than RETAIN-Co-FT in the items-to-basket environment, RETAIN-Co-FT does significantly better in the other two environments, and performs better on average, as shown in Fig 7. It’s also worth mentioning that this performance boost is much more significant in the real-world DROID experiments.
>
> > Q3 & Q4: Sec. 6.5: How was the $\alpha$ parameter selected? Which $\alpha$ parameter did you use for the DROID experiments? How was it selected? Also, did you use different s for the modalities in the experiments, and how did you select this weighting?
>
> In the original submission, we selected the hyperparameter $\alpha$, the merging coefficient, on one scene based on the best OOD performance. As a reminder, for each task that we consider (e.g. mugs-on-plate in LIBERO, plates in DROID), we test policy performance on several scenes, which are different variations of the same task with different objects, distractors, and backgrounds etc. We originally counted the “tuning OOD scene” as a “test” scene and reported performance based on it, along with the other OOD scenes. However, we have now updated the paper to only count it as a validation scene, and report performance (Fig 6, 7, 10, 11) on test scenes where we do not tune this $\alpha$ parameter.
>
> To reiterate, the updated hyperparameter selection process is: we **select $\alpha$ based on its OOD evaluation performance on a held out validation scene, and then we use the same $\alpha$ to report performance on test scenes**. We also added two new test scenes per task for our DROID experiments (see Fig 5 for updated visuals), and we want to highlight that our previously observed trends still hold, and do not change our experimental conclusions. While we do pick $\alpha$ differently per task, which is the common practice for hyperparameter tuning in ML literature, we actually find that $\alpha$ values can be robust to different tasks (see our discussion below on DROID experiments).
>
> Next, to answer your question specifically about the hyperparameter selection on DROID experiments: We only test three values of $\alpha$: 0.25, 0.5, 0.75 for each task, and find that 0.5 works best almost across all tasks and methods (RETAIN-task-FT and RETAIN-Co-FT), with the exception of one. This shows that the $\alpha$ value is quite robust for different tasks. For DROID experiments, we do not test the modality-specific merging, and we merge all parameters based on a single $\alpha$ value. We have updated the details in Appendix A.6.
>
> > Q5: How does alpha depend on training parameters (how many updates) etc?
>
> We did not tune the $\alpha$ parameter based on the number of gradient steps. We pick gradient steps according to how well the fine-tuned model fits the data (measured by ID evaluation performance), and pick $\alpha$ according to the process described above. In general, as described above, we find $\alpha$ to be quite robust.
>
> > Related work: also a paper that looks into finetuning from few demonstrations: Active Fine-Tuning of Multi-Task Policies, Bagatella et al This paper does not do model merging but learned output merging and might be interesting to relate to.
>
> Thanks for pointing this out! We have added discussion of this in the Related Works section.

---

### Author Response · Authors · 2025-11-21
**General Overview for All Reviewers**

Thank you to all the reviewers for your feedback and comments! We have updated the paper (edits highlighted in red) with additional simulated and real world robot experiments, and added more analysis to shed light into why RETAIN is simple yet effective. In summary, we (1) added *four new DROID OOD scenes* for real robot evaluation (results in Fig 6 & 10, visual in Fig 5) and find that our method RETAIN significantly outperform baselines, (2) added analysis and ablation experiments to understand overfitting and the parameter trajectory during fine-tuning, aiming to shed light on why model parameter merging is an effective method for out-of-distribution generalization, and (3) added additional discussion on how we chose hyperparameters such as the merging coefficient and the number of gradient steps.

Please let us know if you have any additional concerns and we are happy to discuss!

---

### Meta-Review · Area_Chair_r9iY · 2025-12-03

**Summary:**

The paper proposes RETAIN, a method for fine-tuning generalist vision-language-action robot policies via parameter merging. All four reviewers recognized the practical importance of the problem and appreciated the simplicity and effectiveness of the proposed approach. However, reviewers raised several concerns that informed the evaluation. The primary concerns included: (1) limited theoretical understanding of why parameter merging works in the non-convex loss landscape of deep neural networks (R6dX, Rx6M), (2) unclear hyperparameter selection process, particularly for the merging coefficient, with concerns about generalization to new tasks without extensive tuning (R6dX, mT9g, Rx6M, Jiug), (3) whether the overfitting problem could be addressed through standard hyperparameter optimization such as learning rate and gradient step tuning rather than parameter merging (R6dX, mT9g), (4) potential cherry-picking of results and inconsistent performance across different tasks (R6dX), and (5) insufficient evidence for continual learning claims based on only two sequential tasks (Rx6M).

**Reviewer Concerns:**

The authors provided a rebuttal that effectively addressed most concerns. They added four new DROID out-of-distribution scenes with 240 evaluation trials, demonstrating RETAIN significantly outperforms baselines without additional tuning. New ablation studies (Appendices A.2-A.3) showed overfitting persists with careful learning rate and gradient step tuning, and that the fine-tuning trajectory is highly non-linear, explaining why parameter merging achieves better solutions. The merging hyperparameter selection process was clarified with it being selected on validation scenes and tested on separate test scenes. The authors also provided theoretical grounding through connections to linear mode connectivity literature. The cherry-picking concern was resolved by clarifying all results were averaged across tasks. The continual learning evaluation remains limited to two tasks but was strengthened with additional test scenes.

**Reviewer Scores:**

R6dX initially rated 4 (marginally below acceptance) but stated they "would not mind if paper is accepted." Given the thorough responses addressing hyperparameter selection and new experimental results, this reviewer would likely have increased their score.
Reviewer mT9g wuld have liekly increased their score after the authors addressed concerns about overfitting and hyperparameter choices.
Reviewers Jiug and Rx6M would have likely both maintained their scores of 8 after rebuttal.

---

### Decision · Program_Chairs · 2026-01-26

Accept (Poster)